# POPULATION-SIZE-AWARE POLICY OPTIMIZATION FOR MEAN-FIELD GAMES

**Pengdeng Li**[1]  **Xinrun Wang**[1*]  **Shuxin Li**[1]  **Hau Chan**[2]  **Bo An**[1]
[1]Nanyang Technological University, Singapore    [2]University of Nebraska, Lincoln, USA
{pengdeng.li,xinrun.wang,shuxin.li,boan}@ntu.edu.sg, hchan3@unl.edu

## ABSTRACT

In this work, we attempt to bridge the two fields of finite-agent and infinite-agent games, by studying how the optimal policies of agents evolve with the number of agents (population size) in mean-field games, an agent-centric perspective in contrast to the existing works focusing typically on the convergence of the empirical distribution of the population. To this end, the premise is to obtain the optimal policies of a set of finite-agent games with different population sizes. However, either deriving the closed-form solution for each game is theoretically intractable, training a distinct policy for each game is computationally intensive, or directly applying the policy trained in a game to other games is sub-optimal. We address these challenges through the **P**opulation-size-**A**ware **P**olicy **O**ptimization (PAPO). Our contributions are three-fold. First, to efficiently generate efficient policies for games with different population sizes, we propose PAPO, which unifies two natural options (augmentation and hypernetwork) and achieves significantly better performance. PAPO consists of three components: i) the population-size encoding which transforms the original value of population size to an equivalent encoding to avoid training collapse, ii) a hypernetwork to generate a distinct policy for each game conditioned on the population size, and iii) the population size as an additional input to the generated policy. Next, we construct a multi-task-based training procedure to efficiently train the neural networks of PAPO by sampling data from multiple games with different population sizes. Finally, extensive experiments on multiple environments show the significant superiority of PAPO over baselines, and the analysis of the evolution of the generated policies further deepens our understanding of the two fields of finite-agent and infinite-agent games.

## 1 INTRODUCTION

Games involving a finite number of agents have been extensively investigated, ranging from board games such as Go (Silver et al., 2016; 2018), Poker (Brown & Sandholm, 2018; 2019; Moravčík et al., 2017), and Chess (Campbell et al., 2002) to real-time strategy games such as StarCraft II (Vinyals et al., 2019) and Dota 2 (Berner et al., 2019). However, existing works are typically limited to a handful of agents, which hinders them from broader applications. To break the curse of many agents (Wang et al., 2020), mean-field game (MFG) (Huang et al., 2006; Lasry & Lions, 2007) was introduced to study the games that involve an infinite number of agents. Recently, benefiting from reinforcement learning (RL) (Sutton & Barto, 2018) and deep RL (Lillicrap et al., 2016; Mnih et al., 2015), MFG provides a versatile framework to model games with large population of agents (Cui & Koeppl, 2022; Fu et al., 2019; Guo et al., 2019; Laurière et al., 2022; Perolat et al., 2021; Perrin et al., 2022; 2020; Yang et al., 2018).

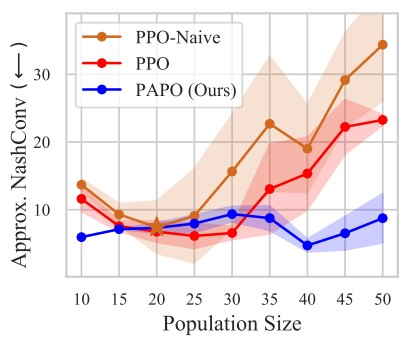

Figure 1: Experiments on Taxi Matching environment show the failure of two naive methods and the success of our PAPO. ↓ means the lower the better performance. See Sec. 5.1 for details.

---

*Corresponding author.

Despite the successes of finite-agent games and infinite-agent games, the two fields are largely evolving independently. Establishing the connection between an MFG and the corresponding finite-agent Markov (stochastic) games[1] has been a research hotspot and it is done typically by the convergence of the empirical distribution of the population to the mean-field (Saldi et al., 2018; Cui & Koeppl, 2022; Cui et al., 2022; Fabian et al., 2022). However, few results have been achieved from an agent-centric perspective. Specifically, a fundamental question is: how do the optimal policies of agents evolve with the population size? As the population size increases, the finite-agent games approximate, though never equal to, their infinite-agent counterparts (Cui & Koeppl, 2021; Mguni et al., 2018). Therefore, the solutions returned by methods in finite-agent games should be consistent with that returned by methods in infinite-agent games. As we can never generate finite-agent games with infinite number of agents, we need to investigate the evolution of the optimal policies of agents, i.e., scaling laws[2], to check the consistency of the methods. However, theoretically investigating the scaling laws is infeasible, as obtaining the closed-form solutions of a set of finite-agent games is typically intractable except for some special cases (Guo & Xu, 2019). Hence, another natural question is: how to efficiently generate efficient policies for a set of finite-agent games with different population sizes? Most methods in finite-agent games can only return the solution of the game with a given number of agents (Bai & Jin, 2020; Jia et al., 2019; Littman et al., 2001). Unfortunately, the number of agents varies dramatically and rapidly in many real-world scenarios. For example, in Taxi Matching environment (Nguyen et al., 2018; Alonso-Mora et al., 2017), the number of taxis could be several hundred in rush hour while it could be a handful at midnight. Fig. 1 demonstrates the failure of two naive options in this environment: i) directly apply the policy trained in a given population size to other population sizes (PPO-Naive), and ii) train a policy by using the data sampled from multiple games with different population sizes (PPO). Furthermore, computing the optimal policies for games with different population sizes is computationally intensive.

In this work, we propose a novel approach to efficiently generate efficient policies for games with different population sizes, and then investigate the scaling laws of the generated policies. Our main contributions are three-fold. First, we propose PAPO, which unifies two natural methods: augmentation and hypernetwork, and thus, achieves better performance. Specifically, PAPO consists of three components: i) the population-size encoding which transforms the original value of population size to an equivalent encoding to avoid training collapse, ii) a hypernetwork to generate a distinct policy for each game conditioned on the population size, and iii) the population size as an additional input to the generated policy. Next, to efficiently train the neural networks of PAPO, we construct a multi-task-based training procedure where the networks are trained by using the data sampled from games with different population sizes. Finally, extensive experiments on multiple widely used game environments demonstrate the superiority of PAPO over several naive and strong baselines. Furthermore, with a proper similarity measure (centered kernel alignment (Kornblith et al., 2019)), we show the scaling laws of the policies generated by PAPO, which deepens our understanding of the two fields of finite-agent and infinite-agent games. By establishing the state-of-the-art for bridging the two research fields, we believe that this work contributes to accelerating the research in both fields from a new and unified perspective.

## 2 RELATED WORKS

Our work lies in the intersection of the two research fields: learning in Markov games (MGs) and learning in mean-field games (MFGs). Numerous works have tried to study the connection between an MFG and the corresponding finite-agent MGs from a theoretical or computational viewpoint such as (Saldi et al., 2018; Doncel et al., 2019; Cabannes et al., 2021), to name a few. The general result achieved is either that the empirical distribution of the population converges to the mean-field as the number of players goes to infinity or that the Nash equilibrium (NE) of an MFG is an approximate NE of the finite-agent game for a sufficiently large number of players, under different conditions such as the Lipschitz continuity of reward/cost and transition functions (Saldi et al., 2018; Cui & Koeppl, 2021; 2022; Cui et al., 2022) or/and the convergence of the sequence of step-graphons (Cui & Koeppl, 2022; Cui et al., 2022). Though the advancements in these works provide a theoretical or

---

[1]In this work, we focus on the finite-agent Markov games sharing a similar structure with the MFG, see Sec. 3 and Appendix A.2 for more details and discussion.

[2]We use the term "scaling laws" to refer to the evolution of agents' optimal policies with the population size, which is different from that of (Kaplan et al., 2020). See Appendix A.4 for a more detailed discussion.

computational understanding of the connection between the two fields, very few results have been achieved from an agent-centric perspective. More precisely, we aim to answer a fundamental question: how do the optimal policies of agents evolve with the population size? Toward this direction, the most related work is (Guo & Xu, 2019), which proves the convergence of the NE of the finite-agent game to that of the MFG by directly comparing the NEs. However, it requires the closed-form solutions of both the finite-agent game and the MFG to be computable, where extra conditions such as a convex and symmetric instantaneous cost function and càdlàg (bang-bang) controls are needed. In a more general sense, deep neural networks have been widely adopted to represent the policies of agents due to their powerful expressiveness (Laurière et al., 2022; Perolat et al., 2021; Perrin et al., 2022; Yang et al., 2018). In this sense, to our best knowledge, our work is the first attempt to bridge the two research fields of finite-agent MGs and MFGs. Specifically, we identify the critical challenges (theoretical intractability, computational difficulty, and sub-optimality of direct policy transfer) and propose novel algorithms and training regimes to efficiently generate efficient policies for finite-agent games with different population sizes and then investigate the scaling laws of the policies. For more discussion and related works on the two fields, see Appendices A.2 and A.3.

Our work is also related to Hypernetwork (Ha et al., 2016). A Hypernetwork is a neural network designed to produce the parameters of another network. It has gained popularity recently in a variety of fields such as computer vision (Brock et al., 2018; Jia et al., 2016; Littwin & Wolf, 2019; Potapov et al., 2018) and RL (Huang et al., 2021b; Rashid et al., 2018). In (Sarafian et al., 2021), Hypernetwork was used to map the state or task context to the Q or policy network parameters. Differently, in addition to generating a distinct efficient policy for each population size by using Hypernetwork, we also investigate the scaling laws of the generated policies, which has not been done in the literature.

## 3 PRELIMINARIES

In this section, we present the game models (finite-agent Markov game and infinite-agent mean-field game) and problem statement of this work.

**Finite-agent Markov Game**. A Markov Game (MG) with $N < \infty$ homogeneous agents is denoted as $G(N) = (\mathcal{N}, \mathcal{S}, \mathcal{A}, p, r, \mathcal{T})$. $\mathcal{N} = \{1, \cdots, N\}$ is the set of agents. $\mathcal{S}$ and $\mathcal{A}$ are respectively the shared state and action spaces. At each time step $t \in \mathcal{T} = \{0, 1, \cdots, T\}$, an agent $i$ in state $s_t^i$ takes an action $a_t^i$. Let $z_{\boldsymbol{s}_t}^N$ denote the empirical distribution of the agent states $\boldsymbol{s}_t = (s_t^1, \cdots, s_t^N)$ with $z_{\boldsymbol{s}_t}^N(s) = \frac{1}{N} \sum_{i=1}^N \mathbf{1}\{s_t^i = s\}$, for all $s \in \mathcal{S}$. Then, $z_{\boldsymbol{s}_t}^N \in \Delta(\mathcal{S})$, the probability distribution over $\mathcal{S}$. For agent $i$, given the state $s_t^i$, action $a_t^i$, and state distribution $z_{\boldsymbol{s}_t}^N$, its dynamical behavior is described by the state transition function $p : \mathcal{S} \times \mathcal{A} \times \Delta(\mathcal{S}) \to \Delta(\mathcal{S})$ and it receives a reward of $r(s_t^i, a_t^i, z_{\boldsymbol{s}_t}^N)$. Let $\pi^i : \mathcal{S} \times \mathcal{T} \to \Delta(\mathcal{A})$ be the policy[3] of agent $i$. Accordingly, $\boldsymbol{\pi} = (\pi^i)_{i \in \mathcal{N}}$ is the joint policy of all agents and $\boldsymbol{\pi}^{-i} = (\pi^j)_{j \in \mathcal{N}, j \neq i}$ is the joint policy of all agents except $i$. Given the initial joint state $\boldsymbol{s}$ and joint policy $\boldsymbol{\pi}$, the value function for agent $1 \leq i \leq N$ is given by

$$V^i(s^i, \pi^i, \boldsymbol{\pi}^{-i}) = \mathbb{E}\left[\sum_{t=0}^T r(s_t^i, a_t^i, z_{\boldsymbol{s}_t}^N)\Big| s_0^i = s^i, s_{t+1}^i \sim p, a_t^i \sim \pi^i\right]. \tag{1}$$

A joint policy $\boldsymbol{\pi}^* = (\pi^{i,*})_{i \in \mathcal{N}}$ is called a Nash policy if an agent has no incentive to unilaterally deviate from others: for any $i \in \mathcal{N}$, $V^i(s^i, \boldsymbol{\pi}^*) \geq \max_{\pi^i} V^i(s^i, \pi^i, \boldsymbol{\pi}^{-i,*})$. Given a joint policy $\boldsymbol{\pi}$, the NASHCONV$(\boldsymbol{\pi}) = \sum_{i=1}^N \max_{\hat{\pi}^i} V^i(s^i, \hat{\pi}^i, \boldsymbol{\pi}^{-i}) - V^i(s^i, \boldsymbol{\pi})$ measures the "distance" of $\boldsymbol{\pi}$ to the Nash policy. That is, $\boldsymbol{\pi}$ is a Nash policy if NASHCONV$(\boldsymbol{\pi}) = 0$.

**Mean-Field Game**. A Mean-Field Game (MFG) consists of the same elements as the finite-agent MG except that $N = \infty$, which is denoted as $G(\infty) = (\mathcal{S}, \mathcal{A}, p, r, \mathcal{T})$. In this case, instead of modeling $N$ separate agents, it models a single representative agent and collapses all other (infinite number of) agents into the mean-field, denoted by $\mu_t \in \Delta(\mathcal{S})$. As the agents are homogeneous, the index $i$ is suppressed. Consider a policy $\pi$ as before. Given some fixed mean-field $(\mu_t)_{t \in \mathcal{T}}$ of the population and initial state $s \sim \mu_0$ of the representative agent, the value function for the agent is

$$V(s, \pi, (\mu_t)_{t \in \mathcal{T}}) = \mathbb{E}\left[\sum_{t=0}^T r(s_t, a_t, \mu_t)\Big| s_0 = s, s_{t+1} \sim p, a_t \sim \pi\right]. \tag{2}$$

---

[3]In experiments, we follow (Laurière et al., 2022) to make the policy dependent on time by concatenating the state with time.

$\pi^*$ is called a Nash policy if the representative agent has no incentive to unilaterally deviate from the population (Perolat et al., 2021; Perrin et al., 2020): $V(s, \pi^*, (\mu_t^*)_{t\in\mathcal{T}}) \geq \max_{\hat{\pi}} V(s, \hat{\pi}, (\mu_t^*)_{t\in\mathcal{T}})$, where $(\mu_t^*)_{t\in\mathcal{T}}$ is the mean-field of the population following $\pi^*$. Given a policy $\pi$, the NashConv is defined as $\text{NASHCONV}(\pi) = \max_{\hat{\pi}} V(s, \hat{\pi}, (\mu_t)_{t\in\mathcal{T}}) - V(s, \pi, (\mu_t)_{t\in\mathcal{T}})$ with $(\mu_t)_{t\in\mathcal{T}}$ being the mean-field of the population following $\pi$. Then, $\pi$ is a Nash policy if $\text{NASHCONV}(\pi) = 0$.

**Problem Statement**. Though numerous advancements have been achieved for the finite-agent MGs and infinite-agent MFGs, the two fields are largely evolving independently (we provide more detailed discussions on the connection between MGs and MFGs in Appendix A.2). Bridging the two fields can contribute to accelerating the research in both fields. There are two closely coupled questions: how do the optimal policies of agents evolve with the population size, i.e., scaling laws, and how to efficiently generate efficient policies for finite-agent MGs with different population sizes? Formally, let $G = \{G(\underline{N}), \cdots, G(N), \cdots, G(\bar{N})\}$ denote a set of MGs, where $\underline{N}$ and $\bar{N}$ denote the minimum and maximum number of agents. Let $\pi_N^*$ denote the Nash policy of a game $G(N)$. Let $\rho(N) = \rho(\pi_N^*, \pi_{N+1}^*)$ denote some measure capturing the difference between the Nash policy of $G(N)$ and that of $G(N+1)$ (see Appendix A.5 for more discussion on $\rho$). Then, one question is how $\rho(N)$ changes with $N$. However, directly investigating the evolution of $\rho(N)$ is infeasible, as we need to obtain the Nash policy $\pi_N^*$ for each game $G(N)$. In addition, directly applying the Nash policy $\pi_N^*$ to a game $G(N')$ with $N' \neq N$ could result in worse (or arbitrarily worse) performance, i.e., large (or arbitrarily large) $\text{NASHCONV}(\pi_N^*)$ for $G(N')$, as shown in Fig. 1. Thus, another question is how to efficiently generate a policy $\pi_N$ that works well for each game $G(N) \in G$, i.e., small $\text{NASHCONV}(\pi_N)$, though it may not be the Nash policy $\pi_N \neq \pi_N^*$. Unfortunately, most existing methods can only return the Nash policy for the game with a given $N$, which hinders them from many real-world applications where the number of agents varies dramatically and rapidly, e.g., the number of taxis could be several hundred in rush hour while it could be a handful at midnight (Alonso-Mora et al., 2017). Furthermore, learning the Nash policies of all the games in $G$ is computationally intensive. Therefore, our objective is to develop novel methods that can efficiently generate efficient policies[4] for the games in $G$ and enable us to investigate the scaling laws.

## 4 POPULATION-SIZE-AWARE POLICY OPTIMIZATION

We start by introducing the basic structure of the policy network of a representative agent. Let $\pi_{\boldsymbol{\theta}}$ denote the agent's policy parameterized by $\boldsymbol{\theta} \in \Theta$. Note that $\boldsymbol{\theta}$ can be any type of parameterization such as direct policy parameterizations (Leonardos et al., 2022) and neural networks (Agarwal et al., 2021). In this work, we use a neural network with three layers to represent the policy, i.e., $\boldsymbol{\theta} = (\boldsymbol{\theta_1}, \boldsymbol{\theta_2}, \boldsymbol{\theta_3})$ with $\boldsymbol{\theta}_l = (\boldsymbol{w}_l, \boldsymbol{b}_l, \boldsymbol{g}_l)$ denoting the vectors of weights ($\boldsymbol{w}_l$), biases ($\boldsymbol{b}_l$), and scaling factors ($\boldsymbol{g}_l$) of the layer $1 \leq l \leq 3$. In practice, $\boldsymbol{\theta}$ is typically trained by using RL algorithms (see Sec. 4.3 for practical implementation). However, as mentioned in the problem statement, either training a distinct policy for each game is computationally intractable or naively applying a policy trained under a given $N$ to other games is sub-optimal (as shown in Fig. 1). To address these challenges, we propose a new policy optimization approach, the **P**opulation-size-**A**ware **P**olicy **O**ptimization (PAPO), which is shown in Fig. 2.

### 4.1 POPULATION-SIZE ENCODING

As PAPO is population-size dependent, one may choose to directly take $N$ as the input (we call it the raw encoding (RE, for short)). However, this might work poorly in practice. In many real-world applications, the population size can be large and vary dramatically and rapidly, e.g., the total number of taxis in Manhattan is around 13,000 and in one day it can fluctuate from a handful at midnight to several hundred in rush hour (Alonso-Mora et al., 2017). When directly taking population sizes as inputs, bigger $N$'s may have a higher contribution to the network's output error and dominate the update of the network. This could degrade the performance or severely, collapse the training (as observed in experiments where the training of PAPO with RE is collapsed).

To address the aforementioned issue, we propose the population-size encoding to pre-process $N$. We use binary encoding (BE, for short) to equivalently transform the original decimal representation of $N$ into a binary vector with size $k > 0$. Formally, given $N$, we obtain a vector $\boldsymbol{e}(N) = [e_1, \cdots, e_k]$

---

[4]We call a policy an "efficient policy" if it has a small NashConv in the corresponding game.

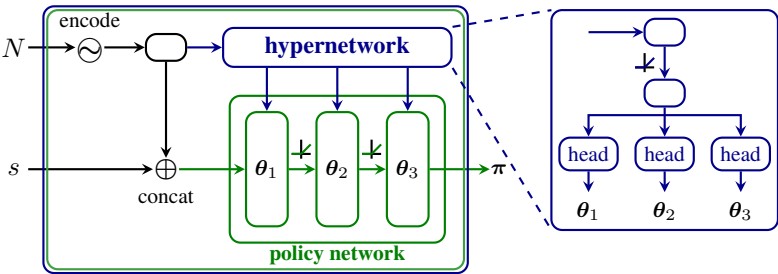

Figure 2: The neural network architecture of PAPO.

such that $N = \sum_{j=1}^{k} 2^{k-j} e_j$ where $e_j \in \{0, 1\}$. After that, the encoding is further mapped to the embedding (through an embedding layer parameterized with $\boldsymbol{\eta}$) which will be fed to the policy network and hypernetwork as shown in Fig. 2. With a slight abuse of notation, throughout this work, we simply use $N$ to represent the embedding of $\boldsymbol{e}(N)$ if it is clear from the context.

## 4.2 POPULATION-SIZE-AWARE ARCHITECTURE

The critical insight of our approach is to leverage the population-size information to generate efficient policies for the games in $G$. In view of the fact that none of the existing works has addressed a similar problem as ours, we first propose two natural methods: augmentation and hypernetwork.

**Augmentation**. That is, we concatenate $N$ with $s_t$ and then pass the resulting input to the policy. A similar idea has been adopted in the literature to address the problem of policy generalization. For example, in (Perrin et al., 2022), the state is concatenated with the initial mean-field to endow the policy with the capability of generalizing across the mean-field space. In our context, by augmenting the state with $N$, the policy can work well across games with different $N$'s.

**Hypernetwork**. That is, we generate a distinct policy for each game by mapping $N$ to the policy network parameters through a hypernetwork (Ha et al., 2016; Sarafian et al., 2021). This choice is completely different from the augmentation as the game-level information (population size in our context) is fully disentangled from the policy input. More specifically, let $h_{\boldsymbol{\beta}}$ denote a hypernetwork parameterized with $\boldsymbol{\beta}$. $h_{\boldsymbol{\beta}}$ takes $N$ as input and outputs a set of parameters (weights, biases, and scaling factors) which will be reshaped to form the structure of the policy network. Formally, we have $\boldsymbol{\theta} = h_{\boldsymbol{\beta}}(N)$. In the hypernetwork, three independent heads are used to generate the parameters of the three layers of the policy network (more details of the network architecture such as the number of neurons of each layer can be found in Appendix B.1).

Although the above two methods can obtain efficient policies for the games in $G$, there is no affirmative answer to the question of which one is better as they achieve the goal from different angles: augmentation uses a fixed set of policy network parameters and takes elements from the cartesian product of two domains ($\mathcal{S}$ and $\{2, 3, \cdots\}$) as inputs while the hypernetwork generates a distinct set of policy network parameters for each game. In this work, instead of trying to answer this question, we propose a unified framework, PAPO, which encompasses the two options as two special cases. Specifically, as shown in Fig. 2, PAPO uses a hypernetwork to generate the parameters of the policy network conditioned on $N$ and also takes $N$ as an additional input to the generated policy.

Intuitively, PAPO preserves the merits of the two special cases and possibly achieves better performance. On one hand, it induces a hierarchical information structure. To a certain extent, it decouples the game-level information from the policy by introducing a hypernetwork to encode the varying elements across different games. In the context of our work, the only difference between the games in $G$ is the population size while other game elements such as the state and action spaces are identical. Thus, PAPO can generate a distinct policy (or in other words, a specialized policy) for each game conditioned on the population size. On the other hand, once the policy is generated, PAPO becomes the augmentation, which, as mentioned before, can work well in the corresponding game.

Furthermore, we can investigate the scaling laws of the generated policies by studying the difference between them. Formally, let $\pi_{\boldsymbol{\theta}=h_{\boldsymbol{\beta}}(N)}$ and $\pi_{\boldsymbol{\theta}=h_{\boldsymbol{\beta}}(N+1)}$ be two policies generated by PAPO for the two games with $N$ and $N+1$ agents, respectively. Then, the difference, denoted as $\rho(N) =$

$\rho\left(\pi_{\boldsymbol{\theta}=h_{\boldsymbol{\beta}}(N)}, \pi_{\boldsymbol{\theta}=h_{\boldsymbol{\beta}}(N+1)}\right)$, as a function of $N$, characterizes the evolution of the policies with the population size. In practice, one can choose any type of $\rho(N)$. In experiments, we will identify a proper measure (a similarity measure) that can be used to achieve this goal.

### 4.3 PRACTICAL IMPLEMENTATION

Notice that PAPO is a general approach. In this section, we specify the practical implementations of the modules in PAPO and construct a training procedure to efficiently train the neural networks.

**Algorithmic Framework.** We implement PAPO following the framework of PPO (Schulman et al., 2017) as it is one of the most popular algorithms for solving decision-making problems. That is, in addition to the policy network $\boldsymbol{\theta}$ (i.e., actor), a value network $\boldsymbol{\phi}$ (i.e., critic) is used to estimate the value of a state and also generated by a hypernetwork. Then, the networks (hypernetworks and embedding layers) are trained with the following loss function (see Appendix B.2 for details):

$$L_t(\boldsymbol{\beta}^{\mathrm{A}}, \boldsymbol{\beta}^{\mathrm{C}}) = \mathbb{E}\left[L_t^1(\boldsymbol{\theta} = h_{\boldsymbol{\beta}^{\mathrm{A}}}(N)) - c_1 L_t^2(\boldsymbol{\phi} = h_{\boldsymbol{\beta}^{\mathrm{C}}}(N)) + c_2 \mathcal{H}(\pi_{\boldsymbol{\theta}=h_{\boldsymbol{\beta}^{\mathrm{A}}}(N)})\right]. \tag{3}$$

As the agents learn their policies independently, the policy learning in this work falls in the category of using independent RL to find Nash equilibrium (Ozdaglar et al., 2021). Recent works (Ding et al., 2022; Fox et al., 2021; Leonardos et al., 2022) have shown that independent policy gradient can converge to the Nash equilibrium under some conditions, which, to some extent, provides support to our work. More discussion can be found in Appendix A.1.

**Training Procedure.** To make PAPO capable of generating efficient policies for the games in the set $G$, a trivial idea is to train the networks sequentially over $G$, i.e., train the networks until they converge in one game and then proceed to the next. Unfortunately, such a method is computationally intensive and could suffer catastrophic forgetting, i.e., the networks focus on the learning in the current game. A more reasonable approach is to train the networks by using the data sampled from all the games (a way similar to multi-task learning (Zhang & Yang, 2021)). Inspired by this idea, we construct a multi-task-based training process to efficiently train the networks. At the beginning of an episode, a game $G(N)$ is uniformly sampled from $G$. Then, PAPO takes $N$ as input and generates a policy which is used by the agents to interact with the environment. Finally, the experience tuples are used to train the networks. More details can be found in Appendix B.3.

## 5 EXPERIMENTS

We first discuss the environments, baselines, metric used to assess the quality of learning, similarity measure used to investigate the scaling laws, and then present the results and ablation studies.

**Environments.** We consider the following environments which have been widely used in previous works: Exploration (Laurière et al., 2022), Taxi Matching (Nguyen et al., 2018), and Crowd in Circle (Perrin et al., 2020). Details of these environments can be found in Appendix C.1.

**Baselines.** (1) PPO: the standard deep RL method. (2) AugPPO: augments the input of PPO with the population size. (3) HyperPPO: without the additional input to the generated policy. (4) PPO-Large: the number of neurons of each layer is increased such that the total number of learnable parameters is similar to PAPO. (5) AugPPO-Large: similar to PPO-Large. The last two baselines are necessary to show that performance gain is not due to the additional number of learnable parameters, but due to the architecture of PAPO. In addition, all the baselines are trained with the same training procedure as PAPO, which ensures a fair comparison between them and PAPO. More details on baselines and hyperparameters are given in Appendix C.2 and Appendix C.3, respectively.

**Approximate NashConv.** Computing the exact NashConv is typically intractable in complex games as it requires obtaining the exact best response (BR) policy of the representative agent. Instead, we obtain an approximate BR by training it for a large enough number of episodes ($1e^6$) and thus, obtain an approximate NashConv, which is the common choice in the literature (Laurière et al., 2022; Perrin et al., 2020). More details are given in Appendix C.4. Furthermore, in Appendix D.2, we show the BR training curves to demonstrate that the BR has approximately converged, ensuring that the approximate NashConv is a reasonable metric to assess the quality of learning.

**Similarity Measure.** Intuitively, two policies are similar means that their output features (representations) of the corresponding layers (not their parameter vectors) are similar, given the same input

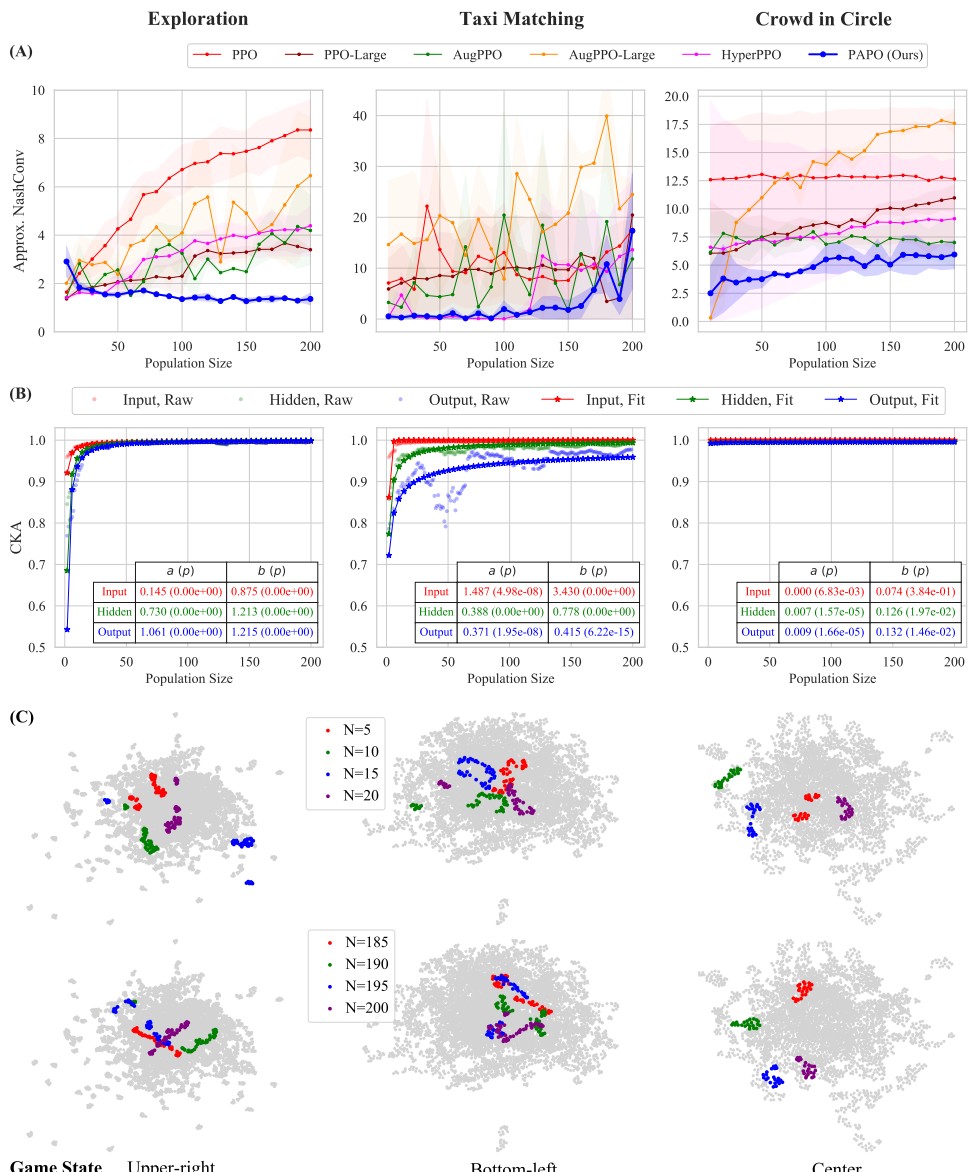

Figure 3: Experimental results. **(A)** Approximate NashConv versus population size. **(B)** CKA value versus population size. **(C)** UMAP embeddings of some human-understandable game states: agent on the upper-right/bottom-left of the map and agent on the center of the circle.

data. Particularly, the outputs of the final layer are closely related to an agent's decision-making (behavior). Therefore, we use the centered kernel alignment (CKA) (Kornblith et al., 2019) to measure the similarity between two policies. More details can be found in Appendix C.5.

## 5.1 RESULTS

In Fig. 1, we show the results obtained in a small-scale setting ($2 \le N \le 50$) in Taxi Matching environment to illustrate our motivation. There are two most straightforward options to obtain policies for different games: i) directly apply the policy trained in a given game ($N = 20$, marked by star) to other games, i.e., PPO-Naive, and ii) train a single policy by using the training procedure in Sec. 4.3 and then apply it to the evaluated games, i.e., PPO. At first glance, it might be natural to expect PPO to work well across different games when it is trained by using the data sampled from

different games. However, as shown in the results, though it outperforms PPO-Naive, it still has a large approximate NashConv when evaluating in different games. In striking contrast, PAPO works well across different games. In addition, given the same training budget, PAPO still preserves a similar performance in the game ($N = 20$) where PPO-Naive was trained.

In Fig. 3, we show the results obtained in a larger-scale setting ($2 \leq N \leq 200$) in different environments. From Panel A, we can draw the following conclusions. (1) PAPO significantly outperforms other naive and strong baselines in terms of approximate NashConv across different games, which demonstrates that PAPO has achieved one of our goals – efficiently generating efficient policies for a set of finite-agent games. (2) As two special cases of PAPO, AugPPO and HyperPPO can work well across different games. However, they are competitive, which verifies that there is no affirmative answer to the question of which one is better (Sec. 4.2 and Appendix D.1). (3) Given the same training budget, simply increasing the number of learnable parameters of the policy (PPO-Large and AugPPO-Large) or only relying on the hypernetwork to generate the policies (Hyper-PPO), though could outperform PPO, is still struggling to achieve better performance. This shows the superiority of PAPO as a unified framework which inherits the merits of the two special cases (AugPPO/AugPPO-Large and HyperPPO) and thus, achieves a better performance.

In Panel B, we show how the policies generated by PAPO change with the population size. From the results we can see that the similarity between two policies increases with the increasing population size. In other words, with the increase in population size, the policies in the games with different population sizes tend to be similar to each other, which shows a convergent evolution of the policies in terms of population size. Though it is impractical to experimentally set $N = \infty$, from the results we hypothesize that the similarity measure increases to 1 when $N \rightarrow \infty$ (the finite-agent game would become an infinite-agent MFG when $N = \infty$). To quantitatively describe the scaling laws, we show the curves fitting the similarity, which is obtained by employing the tools for curve fitting in Numpy (Harris et al., 2020). We consider a function with the form of $\rho(N) = 1 - \frac{a}{N^b} \leq 1$, which maps $N$ to the similarity measure. The fitted curves again verify the aforementioned conclusions. Furthermore, we compute the p-value of each parameter of the function ($a$ and $b$), which shows that the curves fit well ($p < 0.05$) the original CKA values. Notice that there could be different forms of $\rho(N)$. For example, a more general case is $\rho(N) = 1 - \frac{a}{cN^b+d} \leq 1$. However, by computing the p-values, we found that some parameters ($c$ and $d$) are less significant and hence, can be removed from $\rho(N)$. We give a more detailed analysis in Appendix D.3.

In addition, we observed that the variance of the similarity of the input layer is small while that of the output layer is large (especially when $N$ is small). This coincides with the well-known conclusion in deep learning (Alzubaidi et al., 2021; LeCun et al., 2015): the first layer of a neural network extracts the lowest-level features while the last layer extracts the highest-level features. In our context, the only difference between two games is the number of agents while the underlying environmental setup (e.g., grid size) is the same. Hence, the low-level features extracted by the first layers of the policies are similar. In contrast, as the output layers capture the features of agents' decision makings which are largely impacted by the presence of other agents (more precisely, the number of agents), the extracted high-level features could be very different in different games (especially for small $N$'s).

To gain more intuition on how the generated policies are different from each other, in Panel C, we perform an analysis of the policies' output representations similar to (Jaderberg et al., 2019). We manually construct some high-level game states such as "agent on the upper-right/bottom-left of the map" and "agent on the center of the circle" and obtain the output representations of the policy of each game $G(N)$ by using UMAP (McInnes et al., 2018). We colored the representations of some policies: $N \in \{5, 10, 15, 20\}$ and $N \in \{185, 190, 195, 200\}$ to represent small-scale and large-scale settings, respectively. From the results, we can see that when $N$ is large, the policies of different games tend to encode the same game states in similar ways (i.e., the UMAP embeddings of the same game state of different policies tend to be centered), which is aligned with the findings revealed by the CKA values. We give more analysis in Appendix D.4.

## 5.2 ABLATIONS

To gain a deeper understanding of the proposed approach and further support the results obtained in the previous section, we provide more discussion in this section and Appendix D.

**Effect of Population-size Encoding**. In Fig. 4, we show the training curves of PAPO when using RE (i.e., directly using $N$ as the input). It is observed that using RE can result in training collapse. In Taxi Matching environment, though PAPO with RE still gets positive rewards, it is lower than PAPO with BE. The results demonstrate the importance of the population-size encoding for training the neural networks of PAPO. More detailed analysis can be found in Appedix D.5.

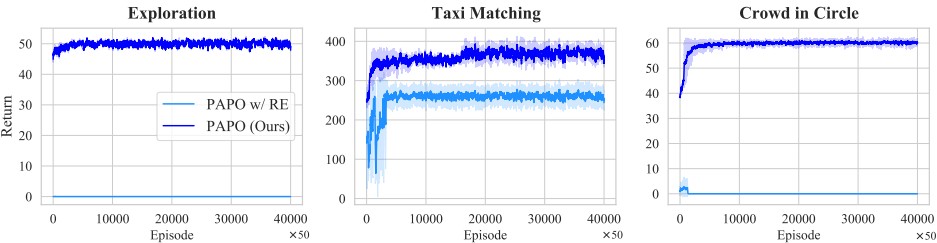

Figure 4: Collapsed training of PAPO when using RE.

**Generalization to Unseen Games**. In Fig. 5, we show the performance of applying PAPO and other baselines to unseen games. We found that, though the population-size aware methods (PAPO, HyperPPO, AugPPO, and AugPPO-Large) could perform better than the population-size unaware methods (PPO and PPO-Large) in some of the unseen games, their performance fluctuates and could be much worse. Therefore, an important future direction is to improve the generalizability of PAPO (as well as other population-size-aware methods), where more advanced techniques such as adversarial training (Ganin et al., 2016) are needed.

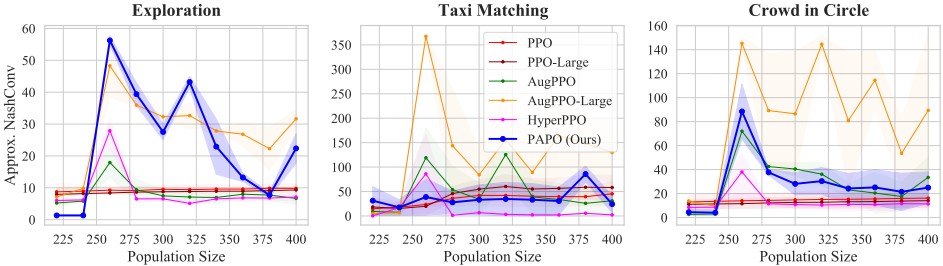

Figure 5: Performance of PAPO and baselines on unseen games.

# 6 CONCLUSIONS AND FUTURE WORKS

In this work, we attempt to bridge the two research fields of finite-agent and infinite-agent games from an agent-centric perspective with three main contributions: i) a novel policy optimization approach, PAPO, to efficiently generate efficient policies for a set of games with different population sizes, ii) a multi-task-based training procedure to efficiently train the neural networks of PAPO, and iii) extensive experiments to demonstrate the significant superiority of our approach over several naive and strong baselines, and the analysis of the scaling laws of the policies to further deepen our understanding of the two research fields of finite-agent and infinite-agent games.

There are several future directions. (1) Beyond the single type of agents, in real-world scenarios, the agents can be divided into multiple types (Ganapathi Subramanian et al., 2020; Yang et al., 2020a), e.g., in Taxi Matching environment, different vehicles have different capacities (Alonso-Mora et al., 2017). Investigating the scaling laws of each type requires new methods and training regimes. (2) Beyond the population size (Gatchel, 2021), in future works, other game elements such as state and action spaces or even the rules of the games can be different between games. PAPO demonstrates the possibility of learning a universal policy for completely different games. (3) Beyond the target set of games, generalization to unseen games requires more advanced techniques such as adversarial training (Ganin et al., 2016). (4) Our approach may have implications for Meta-RL (Fakoor et al., 2019; Finn et al., 2017; Rakelly et al., 2019; Sohn et al., 2019) as well.

## ACKNOWLEDGMENTS

This research is supported by the National Research Foundation, Singapore under its Industry Alignment Fund – Pre-positioning (IAF-PP) Funding Initiative. Any opinions, findings and conclusions or recommendations expressed in this material are those of the author(s) and do not reflect the views of National Research Foundation, Singapore. Hau Chan is supported by the National Institute of General Medical Sciences of the National Institutes of Health [P20GM130461] and the Rural Drug Addiction Research Center at the University of Nebraska-Lincoln. The content is solely the responsibility of the authors and does not necessarily represent the official views of the National Institutes of Health or the University of Nebraska.

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

# A    LEARNING IN MARKOV GAMES

In this section, we provide more discussion on learning in Markov games. First, we present some discussion and analysis on independent learning for finding Nash equilibria of Markov games. Then, we provide more discussion on the connection between MGs and MFGs presented in this work to further facilitate the understanding of our proposal.

## A.1    CONVERGENCE ANALYSIS OF INDEPENDENT POLICY GRADIENT

In a given game, as all the agents learn their policies independently, the policy learning in this work falls in the category of using independent RL to find Nash equilibrium (Ozdaglar et al., 2021). Recent works (Ding et al., 2022; Fox et al., 2021; Leonardos et al., 2022) have shown that independent policy gradient can always converge to the Nash equilibrium under some conditions. In this section, we provide more analysis on the convergence of independent policy gradient in MGs, which, to some extent, provides support to our approach.

### A.1.1    MARKOV POTENTIAL GAMES

In the standard definition of the Markov game (Littman, 1994; Shapley, 1953; Ozdaglar et al., 2021; Ding et al., 2022; Fox et al., 2021; Leonardos et al., 2022), agents are typically assumed to have access to the global state. In the context of our work, the global state is the joint state of all agents $s_t = (s_t^1, \cdots, s_t^N) \in \mathcal{S}^N$. In this sense, a Markov game $G(N)$ is called a Markov potential game (MPG) if there exists a (global) state-dependent potential function $\Phi_s : \Pi \to \mathbb{R}$ such that

$$\Phi_s(\pi^i, \boldsymbol{\pi}^{-i}) - \Phi_s(\hat{\pi}^i, \boldsymbol{\pi}^{-i}) = V^i(s, \pi^i, \boldsymbol{\pi}^{-i}) - V^i(s, \hat{\pi}^i, \boldsymbol{\pi}^{-i}), \tag{4}$$

for all agents $i \in \mathcal{N}$, states $s \in \mathcal{S}^N$ and policies $\pi^i, \hat{\pi}^i \in \Pi^i$, $\boldsymbol{\pi}^{-i} \in \Pi^{-i}$. $\Pi^i$ is the policy space of agent $i$ and $\Pi = \times_{i \in \mathcal{N}} \Pi^i$ ($\Pi^{-i} = \times_{j \in \mathcal{N}, j \neq i} \Pi^j$) is the space of joint policy of all agents (except $i$).

As one of the most important and widely studied classes of games, MPGs have gained much research attention recently due to their power in modeling mixed cooperative/competitive environments and the convergence guarantee of independent policy gradient in these games (Ding et al., 2022; Fox et al., 2021; Leonardos et al., 2022).

### A.1.2    INDEPENDENT POLICY GRADIENT

Assume that all agents update their policies according to the gradient ascent (GA) on their policies independently, i.e., only local information such as each agent's own rewards, actions, and observations (the agent's local state $s \in \mathcal{S}$) is used during the learning process (Leonardos et al., 2022). The GA is given by

$$\pi^{i,k+1} := \pi^{i,k} + \delta \nabla_{\pi^i} V^i(\pi^{i,k}), \quad \forall i \in \mathcal{N}, \tag{5}$$

where $k$ indicates the $k$-th iteration of the policy. Note that the policy $\pi^{i,k+1}$ after $k$-th update is always bounded on its policy space $\pi^{i,k+1} \in \Pi^i$, i.e., $\pi^{i,k+1}(\cdot|s) \in \Delta(\mathcal{A})$ for all $s \in \mathcal{S}$. If not so, a projection operation can be applied to ensure this condition. We omit such an operation for simplicity. Furthermore, we assume that all agents $i \in \mathcal{N}$ use the direct policy parameterization with $\alpha$-greedy exploration as follows:

$$\pi^i(a|s) = (1 - \alpha)x_{i,s,a} + \frac{\alpha}{|\mathcal{A}|}, \tag{6}$$

where $x_{i,s,a} \geq 0$ for all states $s \in \mathcal{S}$, actions $a \in \mathcal{A}$ and $\sum_{a \in \mathcal{A}} \pi^i(a|s) = 1$ for all $s \in \mathcal{S}$. Essentially, $\pi^i(s) = (\pi^i(a|s))_{a \in \mathcal{A}}$ is a mixed strategy in state $s$. $\alpha$ is the exploration parameter.

In practice, the exact gradient $\nabla_{\pi^i} V^i(\pi^{i,k})$ is typically unavailable, agents use stochastic gradient ascent (SGA) to update their policies. Specifically, the exact gradient $\nabla_{\pi^i} V^i(\pi^{i,k})$ is replaced by an estimator, denoted as $\hat{\nabla}_{\pi^i}^k$, which is typically derived from a batch of observations collected through interactions with the game environment by employing the policy $\pi^{i,k}$ at $k$-th iteration. The commonly used (REINFORCE) gradient estimator is as follows:

$$\hat{\nabla}_{\pi^i}^k = R^i \sum_{t=0}^{T} \nabla \log \pi^{i,k}(a_t|s_t), \tag{7}$$

where $R^i = \sum_{t=0}^{T} r_t$ is the sum of rewards of agent $i$ along the trajectory collected by using the policy $\pi^{i,k}$ at $k$-th iteration. Now the SGA is given by

$$\pi^{i,k+1} := \pi^{i,k} + \delta \hat{\nabla}_{\pi^i}^k, \quad \forall i \in \mathcal{N}. \tag{8}$$

### A.1.3 CONVERGENCE ANALYSIS

In this section, we briefly discuss the convergence of independent policy gradient in MGs, following the results obtained in (Leonardos et al., 2022).

**Proposition A.1.1.** *If the reward function is a global signal, i.e., $r(s^i, a^i, z_{\boldsymbol{s}}^N) = r(z_{\boldsymbol{s}}^N)$, $\forall i \in \mathcal{N}$, $\forall s^i \in \mathcal{S}$, and $\forall a^i \in \mathcal{A}$, the finite-agent MG $G(N)$ is an MPG.*

*Proof.* The result can be derived by showing that the conditions in Proposition 3.2 in (Leonardos et al., 2022) are satisfied. Specifically, as the agents share a global reward $r(z_{\boldsymbol{s}}^N)$, $G(N)$ is a potential game at each (global) state $\boldsymbol{s} \in \mathcal{S}^N$, i.e., the potential function is the global reward function $\phi_{\boldsymbol{s}}(\boldsymbol{a}) = r(z_{\boldsymbol{s}}^N)$. Therefore, each agent $i$'s instantaneous reward is decomposed as $r(s^i, a^i, z_{\boldsymbol{s}}^N) = \phi_{\boldsymbol{s}}(\boldsymbol{a}) + u_{\boldsymbol{s}}^i(\boldsymbol{a}^{-i})$, where the dummy $u_{\boldsymbol{s}}^i(\boldsymbol{a}^{-i}) \equiv 0$ trivially satisfies the condition:

$$\nabla_{\pi^i} \mathbb{E}_{\tau \sim \boldsymbol{\pi}} \left[ \sum_{t=0}^{T} u_{\boldsymbol{s}_t}^i(\boldsymbol{a}_t^{-i}) \Big| \boldsymbol{s}_0 = \boldsymbol{s} \right] = (c_{\boldsymbol{s}} \mathbf{1})_{\boldsymbol{s} \in \mathcal{S}^N}, \tag{9}$$

where $\boldsymbol{a}$ and $\boldsymbol{a}^{-i}$ are respectively joint actions of all agents and of all agents other than $i$, $c_{\boldsymbol{s}} \in \mathbb{R}$, $\mathbf{1} \in \mathbb{R}^{|\mathcal{A}|}$, $\tau$ is the trajectory when all agents follow the policy $\boldsymbol{\pi}$. $\square$

Note that here we only give a simple discussion on the connection between our work and the recent advances in the convergence of independent policy gradient, which is not the focus of this work. Indeed, a global reward can ensure that $G(N)$ is an MPG, but we do not impose such a condition in the definition of the game model in Sec. 3. In different benchmark environments, an agent can receive a global reward, a local reward (Laurière et al., 2022), or both (Nguyen et al., 2018).

**Proposition A.1.2.** *(Convergence of Independent Policy Gradient, Theorem 1.2 in (Leonardos et al., 2022)) Consider the MG $G(N)$ satisfying the condition in Proposition A.1.1. If each agent runs SGA 8 using direct policy parameterization (Eq. 6) on their policies and the updates are simultaneous, then the learning converges to an approximate Nash policy.*

Note that the above convergence is established under the direct policy parameterization of $\pi_{\boldsymbol{\theta}}$. In practice, such a choice is typically less expressive than using a neural network to represent the policy. However, establishing the convergence guarantee for neural network-type parameterization is extremely hard, if not impossible, because neural networks are typically highly non-linear and non-convex/concave. Nonetheless, empirical verification can be conducted by computing the (approximate) NashConv of a trained policy $\pi_{\boldsymbol{\theta}}$, i.e., if the (approximate) NashConv is 0, then the policy $\pi_{\boldsymbol{\theta}}$ is a Nash policy, regardless of the type of parameterization.

In our experiments, we use neural networks to represent the policies of agents. In this context, as the (approximate) NashConv measures the distance of a policy to the Nash policy, we train/generate a policy that has a lower (approximate) NashConv, which shows the potential (though not exact) convergence of our approach.

### A.2 CONNECTION BETWEEN MG AND MFG

To facilitate the understanding of our proposal, we give some remarks on the connection between the MG and MFG described in Sec. 3.

We attempt to bridge the two research fields (finite-agent MG and infinite-agent MFG) from a finite-to-infinite perspective. That is, we try to envision how will the policy of an MFG behave by investigating a series of finite-agent MGs. In this sense, the finite-agent MGs we considered should share a similar structure with the MFG, i.e., all agents are homogeneous. Though such game-level connection between MG and MFG is somewhat implicit, MFG provides us the guidance to generate the "correct" finite-agent MGs, i.e., the games that are different only in the number of agents while other elements such as state and action spaces and reward functions are kept identical. Notice that

directly generating a game with an infinite number of agents is impractical as it requires infinite computing resources for simulating the behaviors of an infinite number of agents.

On the policy-level connection, we note that it is not directly comparable between our proposed PAPO and the methods for MFGs. In view of the finite-to-infinite perspective, to implement a fair comparison, naturally we first need to generate a game with an infinite number of agents and then train the networks of PAPO by using the data sampled from this game. However, we can neither generate such a game as it requires infinite computing resources for simulating the behaviors of an infinite number of agents nor train the networks as PAPO will need to take $N = \infty$ as input. In this sense, our experiments do not (or more precisely, cannot) implement such a comparison but provide insights into the relationship between the policies generated by PAPO and the policy of the MFG.

On the other hand, much also can be done from an infinite-to-finite perspective that is opposite to ours. This is grounded on the well-known fact that a policy that gives a mean field equilibrium also gives an $\epsilon(N)$-Nash equilibrium for the corresponding game with $N$ agents, where $\epsilon(N)$ goes to 0 as $N$ goes to infinity (Saldi et al., 2018). In this sense, one can first learn a policy in the MFG (under the assumption that the algorithm has access to a simulator which takes the current state, action, and mean-field as inputs and outputs the reward, next state, and next mean-field (Guo et al., 2019)) and then apply it to the finite-agent games with different $N$'s. Essentially, this is analog to the PPO-Naive mentioned in the Introduction. Differently, here the policy is trained in the MFG while in PPO-Naive it is trained in a finite-agent MG. As a consequence, the (approximate) NashConv could be arbitrarily large when $N$ decreases from large to small (ideally, from $N = \infty$ to $N = 2$). In this sense, it is worth interest to propose novel methods to mitigate the loss of applying the policy learned in the MFG to finite-agent MGs, which we leave for future works.

### A.3 MORE RELATED WORKS

**Learning in Markov Games (MGs)**. Among finite-agent games, Markov game (Littman, 1994; Shapley, 1953) is a widely used framework for characterizing games with sequential decision-making, e.g., in multi-agent reinforcement learning. There is a long line of work in developing efficient algorithms for finding Nash equilibria of MGs under various assumptions such as full environmental knowledge (Hansen et al., 2013; Hu & Wellman, 2003; Littman et al., 2001; Wei et al., 2020), access to simulators (Jia et al., 2019; Sidford et al., 2020; Wei et al., 2017; 2021), and special structures like zero-sum games (Bai & Jin, 2020; Bai et al., 2020; Chen et al., 2021; Huang et al., 2021a; Jin et al., 2021; Xie et al., 2020; Zhang et al., 2020) and potential games (Ding et al., 2022; Fox et al., 2021; Leonardos et al., 2022), or built on techniques for learning single-agent Markov decision process (Bai et al., 2021; Liu et al., 2021). Our work adds to the vast body of existing works on learning in MGs. Instead of concentrating on the policy learning in a given MG, we aim to investigate how the optimal policies of agents evolve with the population size, i.e., scaling laws (Kaplan et al., 2020; Kello et al., 2010; Lobacheva et al., 2020), and propose novel methods to efficiently generate policies that work well across games with different population sizes, given that the agents are homogeneous which is a common phenomenon in many real-world domains such as crowd modeling (Yang et al., 2020b), Ad auction (Gummadi et al., 2012), fleet management (Lin et al., 2018), and sharing economy (Hamari et al., 2016).

**Learning in Mean-Field Games (MFGs)**. In contrast to finite-agent MGs, MFGs consider the case with an infinite number of agents. Since introduced in (Huang et al., 2006) and (Lasry & Lions, 2007), MFG has gained intensive research attention due to its power in modeling games involving a large population of agents (Achdou & Laurière, 2020; Gomes et al., 2014; Lauriere, 2021; Ruthotto et al., 2020). Recently, the capability of MFG is further revealed by benefiting from RL (Fu et al., 2019; Guo et al., 2019; Perolat et al., 2021; Perrin et al., 2020) and deep RL (Laurière et al., 2022; Perrin et al., 2022; Subramanian & Mahajan, 2019; Yang et al., 2018). Despite the progress, the evolution of finite-agent games to infinite-agent MFGs is poorly understood and unexplored when the policies of agents are represented by deep neural networks. In this sense, our work makes the first attempt to bridge the two research fields by establishing a novel tool which we call PAPO.

### A.4 MORE DISCUSSION ON THE TERM "SCALING LAWS"

The terminology "Scaling Law" has been widely used in different areas including biology, physics, social science, and computer science. It typically describes the functional relationship between two

quantities. In this sense, we note that the two quantities are typically problem-dependent, e.g., the performance of a model and the model size (Kaplan et al., 2020), the fluctuations in the number of messages sent by members in a communication network and their level of activity (Rybski et al., 2009), the probability that a vertex in a social network interacts with other vertices and the $k$ other vertices (Barabási & Albert, 1999), to name a few.

In this work, the two quantities are: (1) the behavior of the policy network, and (2) the number of agents. In fact, this is conceptually similar to the scaling laws in social networks which investigate how some quantity (e.g., the property of a social network such as the connectivity) changes with the number of vertices (typically, a vertex stands for an individual, i.e., an agent) (Barabási & Albert, 1999). Our work follows a similar idea but focuses on investigating how the behavior of the policy network change with the number of agents. Therefore, the term "Scaling Law" is suited to this work as, in a more general sense, it can be used to describe the relationship between any two quantities which are determined by the specific problem at hand, not only the size of the model or training set or the amount of compute used for training (Kaplan et al., 2020). Furthermore, in contrast to the areas such as natural language processing (NLP), computer vision (CV), and single-agent RL which typically consider a single model, it is natural to consider the scaling laws of the policy with the number of agents in multi-agent systems (MAS).

### A.5 MORE DISCUSSION ON SIMILARITY MEASURE

In this section, we give more discussion on the similarity measure employed in this work. First, we answer an important question as follows:

*What is the most appropriate measure for studying the scaling laws in the context of this work?*

In this work, we aim to investigate how the Nash policy $\pi_N^*$ changes with the population size $N$. However, it is non-trivial to determine an appropriate measure to capture the evolution of $\pi_N^*$ with $N$. There are two intuitive choices:

(1) The performance of $\pi_N^*$, which is similar to (Kaplan et al., 2020). However, we note that taking the performance as the measure is suitable only when the underlying task is identical, as in (Kaplan et al., 2020). In our work, the games with different $N$'s are essentially different tasks, which means that the performance of (approximate NashConv) is not an appropriate measure. This is also the difference between our work and some machine learning (ML) works such as (Kaplan et al., 2020).

(2) The difference between $\pi_N^*$ and a fixed reference policy $\hat{\pi}$. Such a measure may unable to show how $\pi_N^*$ changes with $N$, because all $\pi_N^*$'s could have the same difference. For example, suppose that $\pi_N^*$ and $\hat{\pi}$ have two parameters (2 dimensions). Then, when all $\pi_N^*$'s form a circle and the centroid is the fixed reference policy $\hat{\pi}$, all $\pi_N^*$'s have the same distance to $\hat{\pi}$, but they are different themselves.

Thus, considering an "absolute" quantity (performance of $\pi_N^*$ or difference between $\pi_N^*$ and a fixed reference policy $\hat{\pi}$) could be struggling in studying the scaling laws of the Nash policies $\pi_N^*$. Instead, we consider the "relative" change between the Nash policies, i.e., the difference $\rho(N)$ between two Nash policies. Intuitively, such a measure could have more implications and does not cause inconsistency with our objective (i.e., how $\pi_N^*$ itself changes with $N$) as one could recover an "absolute" quantity from the "relative" changes. For example, let the reference policy be $\hat{\pi} = \pi_{N+2}^*$, $\rho(N)$ be the difference between $\pi_N^*$ and $\pi_{N+1}^*$, $\rho(N+1)$ be the difference between $\pi_{N+1}^*$ and $\hat{\pi}$. Then, it could be possible to derive the difference between $\pi_N^*$ and $\hat{\pi}$ by combining $\rho(N)$ and $\rho(N+1)$.

In addition, when considering the existence of multiple equilibria, rigorously defining the similarity measure $\rho(N)$ could be more involved and outside the scope of this work, as it could be closely related to closed-form solutions which are typically hard to obtain in complex multi-player games, or needs more elaborate discussion. Nevertheless, in this work, as the policies are represented by deep neural networks (DNNs), we use CKA to measure the similarity between two (approximate) Nash policies (the more similar the smaller difference between them), which is well-defined because: (1) CKA is one of the commonly used measures to characterize the similarity between two DNNs, as given in Appendix C.5 and (Kornblith et al., 2019), and (2) As mentioned in Sec. 5, such a measure is capable of characterizing the intuition: two policies are similar means that their output features (representations) of the corresponding layers (not their parameter vectors) are similar, given the same input data.

## B   MORE DETAILS ON PAPO

In this section, we provide more details on PAPO and practical implementations.

### B.1   NETWORK ARCHITECTURE

The embedding layer is a linear layer with 128 neurons. The policy network consists of three fully-connected (FC) layers, each with 128 neurons. The activation between layers is ReLU.

The hypernetwork starts with two FC layers, each with 128 neurons. The activation between layers is ReLU. After that, three independent heads are used to generate the parameters of the three layers of the policy network. In each head, three linear layers are used to map the output $z$ of the two FC layers to the weights, biases, and scaling factors. Such an architecture is similar to (Sarafian et al., 2021; Littwin & Wolf, 2019). Fig. 6 shows the architecture of the described hypernetwork. The generated parameters are reshaped to form the structure of the policy network. Take the hidden layer ($l = 2$) of the policy network as an example. Suppose that the output from the input layer is $x$ and the weights, biases, and scaling factors generated by the head corresponding to the hidden layer are $\boldsymbol{w}_2(z)$, $\boldsymbol{b}_2(z)$, and $\boldsymbol{g}_2(z)$, respectively. Then, the output to the next layer is $x' = \text{ReLU}(x\boldsymbol{w}_2(z) \odot (1 + \boldsymbol{g}_2(z)) + \boldsymbol{b}_2(z))$. To be more specific, in Fig. 7 we present the code snippet in the forward process of PAPO. The "emb" denotes the embedding of the population size and the batch size is the number of experience tuples (transitions) obtained in every $E$ episodes as the common practice in PPO (see Algorithm 1 for details). The final logits will be used to compute the loss during training or infer the action after passing through a softmax activation during execution.

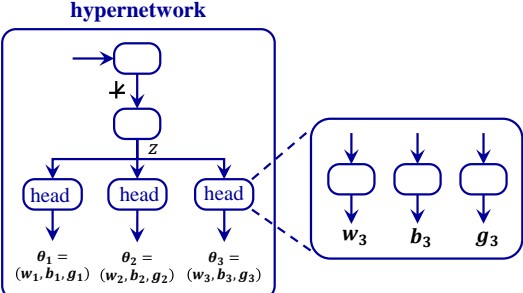

Figure 6: The architecture of hypernetwork.

```
# state:      [batch_size, state_dim]
# emb:        [batch_size, emb_dim]
# w1:         [batch_size, 128, state_dim + emb_dim]
# b1:         [batch_size, 128, 1]
# g1:         [batch_size, 128, 1]
# w2:         [batch_size, 128, 128]
# b2:         [batch_size, 128, 1]
# g2:         [batch_size, 128, 1]
# w3:         [batch_size, action_dim, 128]
# b3:         [batch_size, action_dim, 1]
# g3:         [batch_size, action_dim, 1]

out = F.relu(torch.bmm(w1, torch.cat([state, emb], dim=1).unsqueeze(2)) * (1 + g1) + b1)
out = F.relu(torch.bmm(w2, out) * (1 + g2) + b2)
out = torch.bmm(w3, out) * (1 + g3) + b3
logits = torch.squeeze(out, dim=2)      # logits: [batch_size, action_dim]
```

Figure 7: The code snippet in the forward process of PAPO.

### B.2   LOSS FUNCTION

Let $\pi_{\boldsymbol{\theta}}$ and $V_{\boldsymbol{\phi}}$ respectively denote a representative agent's actor and critic ($\pi_{\boldsymbol{\theta}_{\text{old}}}$ and $V_{\boldsymbol{\phi}_{\text{old}}}$ respectively denote the old version of actor and critic which are periodically updated by copying from $\boldsymbol{\theta}$ and $\boldsymbol{\phi}$). Similar to PPO (Schulman et al., 2017), the loss function for PAPO is:

$$L_t(\boldsymbol{\beta}^{\text{A}}, \boldsymbol{\beta}^{\text{C}}) = \mathbb{E}\Big[L_t^1(\boldsymbol{\theta} = h_{\boldsymbol{\beta}^{\text{A}}}(N)) - c_1 L_t^2(\boldsymbol{\phi} = h_{\boldsymbol{\beta}^{\text{C}}}(N)) + c_2\mathcal{H}(\pi_{\boldsymbol{\theta}=h_{\boldsymbol{\beta}^{\text{A}}}(N)})\Big], \qquad (10)$$

where $h_{\beta^A}$ and $h_{\beta^C}$ are respectively the hypernetworks for generating the actor and critic networks. For notation simplicity, here we use $\boldsymbol{\theta}_N$ and $\boldsymbol{\phi}_N$ to represent $\boldsymbol{\theta} = h_{\beta^A}(N)$ and $\boldsymbol{\phi} = h_{\beta^C}(N)$, respectively. Then, the three terms in the above equation are:

$$L_t^1(\boldsymbol{\theta} = h_{\beta^A}(N)) = \mathbb{E}\left[\min\left(\frac{\pi_{\boldsymbol{\theta}_N}(a_t|s_t)}{\pi_{\boldsymbol{\theta}_{\mathrm{old}}}(a_t|s_t)}A_t, \mathrm{clip}(\frac{\pi_{\boldsymbol{\theta}_N}(a_t|s_t)}{\pi_{\boldsymbol{\theta}_{\mathrm{old}}}(a_t|s_t)}, 1-\epsilon, 1+\epsilon)A_t\right)\right],$$

$$L_t^2(\boldsymbol{\phi} = h_{\beta^C}(N)) = \left(V_{\boldsymbol{\phi}_N}(s_t) - \sum_{t'=t}^{T} r(s_{t'}, a_{t'}, z_{\boldsymbol{s}_{t'}}^N)\right)^2,$$

$$\mathcal{H}(\pi_{\boldsymbol{\theta} = h_{\beta^A}(N)}(\cdot|s_t)) = -\mathbb{E}_{a_t \sim \pi_{\boldsymbol{\theta}_N}} \log \pi_{\boldsymbol{\theta}_N}(a_t|s_t),$$

where $A_t$ is the truncated version of generalized advantage estimation:

$$A_t = \delta_t + (\gamma\lambda)\delta_{t+1} + \cdots + (\gamma\lambda)^{T-t}\delta_T \text{ with } \delta_t = r_t + \gamma V_{\boldsymbol{\phi}_N}(s_{t+1}) - V_{\boldsymbol{\phi}_N}(s_t),$$

where $r_t = r(s_t, a_t, z_{\boldsymbol{s}_t}^N)$ and $\delta_t$ is the TD error (Sutton & Barto, 2018). The expectation $\mathbb{E}$ is taken on a finite batch of experiences sampled by interacting with the environment.

### B.3 TRAINING PROCEDURE

We construct a training procedure where the networks of PAPO are trained by using the data sampled from all the games in the target set $G$. Specifically, at the beginning of each episode, a game $G(N)$ is uniformly sampled from the target set $G$. Then, PAPO takes $N$ as input and generates a policy which is used by the agents to interact with the environment for $T$ steps. Next, the $T$ experience tuples of the representative agent are stored in the episode buffer $\mathcal{D}$. Finally, every $E$ episodes, we optimize the surrogate loss function $L_t(\beta^A, \beta^C)$ with respect to $\beta^A$ and $\beta^C$ (as well as the embedding layers $\eta^A$ and $\eta^C$) for $K$ epochs with the collected $ET$ experience tuples. As the agents are homogeneous, without loss of generality, we choose $i = 1$ as the representative agent. A more detailed description of the training procedure is shown in Algorithm 1.

---

**Algorithm 1:** Training Procedure

---

**Input:** Hyperparameters
**Output:** Trained PAPO

1 $\mathcal{D} \leftarrow \emptyset$;
2 **for** *episode* $= 1, 2, \cdots$ **do**
3      Sample a game $G(N) \sim \mathrm{Uniform}[G]$;
4      $s_0^i \leftarrow s_0, \forall i \in \mathcal{N}$;
5      Generate a policy $\pi_{\boldsymbol{\theta} = h_{\beta^A}(N)}$;
6      **for** $0 \leq t \leq T$ **do**
7          Sample action $a_t^i \sim \pi_{\boldsymbol{\theta}}(\cdot|s_t^i), \forall i \in \mathcal{N}$;
8          Execute $a_t^i$ in $G(N), \forall i \in \mathcal{N}$;
9          Receive reward $r_t = r(s_t^i, a_t^i, z_{\boldsymbol{s}_t}^N), \forall i \in \mathcal{N}$;
10          Observe new state $s_{t+1}^i, \forall i \in \mathcal{N}$;
11          Store data $\mathcal{D} \leftarrow \mathcal{D} \cup \{(s_t^1, a_t^1, r_t, s_{t+1}^1)\}$;
12      **end**
13      **if** *episode*$\%E = 0$ **then**
14          Update $\beta^A$ and $\beta^C$ (and $\eta^A$ and $\eta^C$) by optimizing $L_t$ for $K$ epochs;
15          $\mathcal{D} \leftarrow \emptyset, \boldsymbol{\theta}_{\mathrm{old}} \leftarrow \boldsymbol{\theta}, \boldsymbol{\phi}_{\mathrm{old}} \leftarrow \boldsymbol{\phi}$;
16      **end**
17 **end**

---

## C  EXPERIMENTAL DETAILS

### C.1  DETAILS ON GAME ENVIRONMENTS

**Exploration** (Laurière et al., 2022). Consider a grid environment with size $M \times M$. The state of an agent is his coordinate $s_t = (x, y)$ and available actions are: left, right, up, down, and stay. The agent cannot walk through the walls on the boundaries. Given $s_t = (x, y)$, $a_t$, and $\mu_t$, the reward for the agent is $r(s_t, a_t, \mu_t) = -\log(\mu_t(s_t))$. In experiments, we set $M = 10$.

**Taxi Matching** (Nguyen et al., 2018). Consider a grid world environment with size $M \times M$. A set of drivers aim to maximize their own revenue by picking orders distributed over the map. As the demand in different zones is varied (e.g., the demand in downtown is higher than that in residential areas), each driver needs to find an optimal roaming policy while taking the competition with other drivers into consideration. The state of a driver is his coordinate $s_t = (x, y)$ and actions are: left, right, up, down, and stay. Given $s_t = (x, y)$, $a_t$, and $\mu_t$, the reward for a driver is $r(s_t, a_t, \mu_t) = -o_{s_t} \log(\mu_t(s_t))$, where $o_{s_t}$ denotes the total order price in state $s_t$. In experiments, we set $M = 10$ and consider that there are $100$ orders each with a reward of $1$ and distributed over the map according to the Gaussian distribution with the mean at the center of the map.

**Crowd in Circle** (Perrin et al., 2020). Consider a 1D environment with $M = 20$ states denoted as $\mathcal{S} = \{1, 2, \cdots, |\mathcal{S}| = 20\}$ and form a circle. The available actions for an agent are: left, right, and stay. At each time step $t$, there is a point of interest (PoI, for short) located at $\bar{s}_t$. Given $s_t$, $a_t$, and $\mu_t$, the reward for the agent is $r(s_t, a_t, \mu_t) = -\log(\mu_t(s_t)) + 5 \times \mathbf{1}\{s_t = \bar{s}_t\}$, i.e., the agent gets a reward of 5 when he is located on the PoI while still favoring social distancing. In experiments, we consider two PoIs during an episode: for $t \leq \frac{T}{2}$, $\bar{s}_t = 5$ while for $t > \frac{T}{2}$, $\bar{s}_t = 15$.

### C.2  ALGORITHM FRAMEWORKS AND REMARKS ON BASELINES

In this section, we first introduce the baselines and then present the details of the algorithm frameworks of PAPO and different baselines.

To facilitate the understanding of the experimental design, we introduce the baseline and give some remarks on them. As the network architecture of PAPO is large (has more learnable parameters), in addition to the standard baselines (PPO and AugPPO), we consider three more baselines: PPO-Large, AugPPO-Large, and HyperPPO (note that HyperPPO has a larger number of learnable parameters than PPO and AugPPO and hence, we regard it as a stronger baseline), which have similar numbers of learnable parameters as PAPO by increasing the number of neurons of the hidden layers of the policy network (PPO-Large and AugPPO-Large) or hypernetwork (HyperPPO). This is critical to ensure a fair comparison and demonstrate the effectiveness of our approach. In Table 1, we give the numbers of learnable parameters of different methods in different environments. As the dimensions of the input and output of different environments are different, the numbers are different (notice that for the same environment, the numbers for different methods should be kept similar). In Table 2, we give the numbers of neurons used in different methods to obtain a similar number of learnable parameters. Note that for HyperPPO and PAPO, we change the two FC layers of the hypernetwork as the policy network structure is fixed.

Although the methods using RE (e.g., AugPPO w/ RE, AugPPO-Large w/ RE, HyperPPO w/ RE) can be considered as the baselines, we note that they could be weaker than those using BE (AugPPO, AugPPO-Large, HyperPPO) due to the possible training collapse when using RE, as shown in Fig. 4 and Appendix D.5. Thus, when presenting the main results obtained in experiments in Fig. 3, we focus on the baselines which use BE, which also ensures a fair comparison between them and PAPO (as well as the fair comparison between these baselines). We study the effect of RE on the performance in the ablation study (Fig. 4 and Appendix D.5).

In Fig. 8, we present the algorithm frameworks of PAPO and different baselines. "emb" denotes the embedding of the population size (see Sec. 4.1 and Appendix B.1).

- In PPO/PPO-Large, the actor (critic) takes the agent's state and current time as inputs and outputs the policy (value). PPO-Large operates in the same manner as PPO except that the number of learnable parameters is similar to PAPO.

- In AugPPO/AugPPO-Large, the actor (critic) takes the agent's state, current time, and the embedding of population size as inputs and outputs the policy (value). AugPPO-Large operates in the same manner as AugPPO except that the number of learnable parameters is similar to PAPO.

- In HyperPPO, we first use the hypernetwork to generate the actor (critic) by taking the embedding of population size as input, then the generated actor (critic) takes the agent's state and current time as inputs and outputs the policy (value).

- In PAPO, we first use the hypernetwork to generate the actor (critic) by taking the embedding of population size as input, then the generated actor (critic) takes the agent's state, current time, and the embedding of population size as inputs and outputs the policy (value).

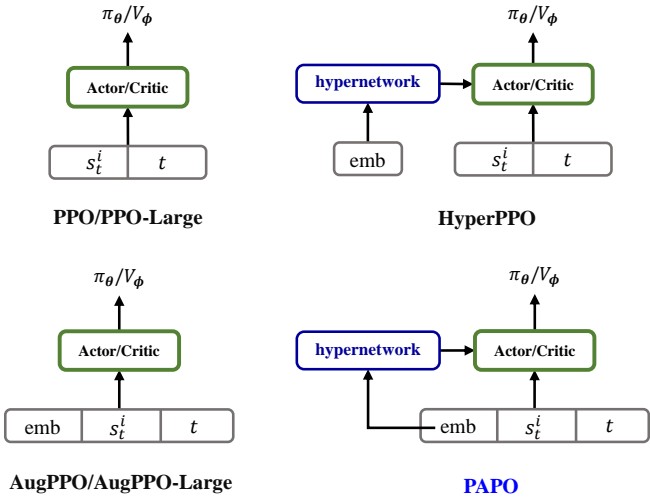

Figure 8: The algorithm frameworks of PAPO and different baselines.

Table 1: The numbers of learnable parameters of different methods in different environments.

| Env./Method | PPO | AugPPO | PPO-Large | AugPPO-Large | HyperPPO | PAPO |
|---|---|---|---|---|---|---|
| Exploration | 45K | 81K | 10,159K | 10,457K | 10,166K | 10,176K |
| Taxi Matching | 44K | 80K | 5,936K | 5,918K | 5,883K | 5,893K |
| Crowd in Circle | 44K | 80K | 10,146K | 10,444K | 9,995K | 10,076K |

Table 2: The numbers of neurons used in different methods.

| Env./Method | PPO-Large | AugPPO-Large | HyperPPO | PAPO |
|---|---|---|---|---|
| Exploration | (2230, 2230, 2230) | (2200, 2200, 2200) | (128, 220) | (128, 128) |
| Taxi Matching | (1700, 1700, 1700) | (1635, 1635, 1635) | (128, 128) | (128, 74) |
| Crowd in Circle | (2230, 2230, 2230) | (2200, 2200, 2200) | (128, 220) | (128, 128) |

## C.3 HYPERPARAMETERS

Table 3 lists the parameters used in our experiments. Unless otherwise specified, the parameters listed in Table 3 are the same in different game environments. The parameters related to the network architectures can be found in Sec. B.1. The parameters related to the environments can be found in the previous section. Without loss of generality, we evaluate the performance of different methods in a subset: $G' = \{10, 20, \cdots, 200\}$. As for generalization to unseen games, we evaluate the methods

in a set of unseen games: $\tilde{G} = \{220, 240, \cdots, 400\}$. The approximate BR policy is obtained by using (single-agent) PPO. Moreover, all experiments are run on a machine with 20 Intel i9-9820X CPUs and 4 NVIDIA RTX2080 Ti GPUs, and averaged over 3 random seeds.

Table 3: Hyperparameters.

| Hyperparameter | Value |
|---|---|
| optimizer | Adam |
| length of an episode $T$ | 20 |
| minimum number of agents $\underline{N}$ | 2 |
| maximum number of agents $\bar{N}$ | 200 |
| maximum number of policy training episodes | $2 \cdot 10^7$ |
| maximum number of BR training episodes | $1 \cdot 10^6$ |
| actor learning rate | $3 \cdot 10^{-5}$ |
| critic learning rate | $3 \cdot 10^{-4}$ |
| update every $E$ episodes | 5 |
| optimize $K$ epochs at each update | 5 |
| critic loss coefficient $c_1$ | 0.5 |
| entropy loss coefficient $c_2$ | 0.01 |
| batch size $m$ for computing CKA | 1000 |
| dimension of binary encoding $k$ | 12 |

## C.4 APPROXIMATE NASHCONV

In this section, we provide more details on how to compute the approximate NashConv. Let $\pi_{\boldsymbol{\theta}_N}$ be the policy returned by a method (PAPO or other baselines) for the given game $G(N)$. According to the definition given in Sec. 3, the NashConv of this policy is as follows:

$$\text{NASHCONV}(\pi_{\boldsymbol{\theta}_N}) = \sum_{i=1}^{N} \max_{\hat{\pi}^i} V^i(s^i, \hat{\pi}^i, \{\pi^j = \pi_{\boldsymbol{\theta}_N}\}_{j=1, j\neq i}^{N}) - V^i(s^i, \{\pi^i = \pi_{\boldsymbol{\theta}_N}\}_{i=1}^{N}). \quad (11)$$

As the agents are homogeneous, without loss of generality, we compute the NashConv of a representative agent $i$. Specifically, we first train an approximate BR policy for agent $i$, denoted as $\pi_{\boldsymbol{\theta}}^{i,\text{BR}}$ (deriving the exact BR policy for the agent $i$ is typically difficult in multi-player games). Then, we compute the approximate NashConv of the agent $i$ as:

$$\text{NASHCONV}^i(\pi_{\boldsymbol{\theta}_N}) = V^i(s^i, \pi_{\boldsymbol{\theta}}^{i,\text{BR}}, \{\pi^j = \pi_{\boldsymbol{\theta}_N}\}_{j=1, j\neq i}^{N}) - V^i(s^i, \{\pi^i = \pi_{\boldsymbol{\theta}_N}\}_{i=1}^{N}). \quad (12)$$

Roughly speaking, the approximate NashConv of agent $i$ is the difference between his value function of following the BR policy $\pi_{\boldsymbol{\theta}}^{i,\text{BR}}$ and his value function of following current policy $\pi_{\boldsymbol{\theta}_N}$, given that the other agents are fixed to the current policy $\{\pi^j = \pi_{\boldsymbol{\theta}_N}\}_{j=1, j\neq i}^{N}$.

## C.5 SIMILARITY MEASURE: CENTERED KERNEL ALIGNMENT

In this work, we use centered kernel alignment (CKA) (Kornblith et al., 2019) to measure the similarity between two policies generated by PAPO. Specifically, we aim to compute the similarity measure $\rho(N) = \rho(\pi_{\boldsymbol{\theta}=h_{\boldsymbol{\beta}}(N)}, \pi_{\boldsymbol{\theta}=h_{\boldsymbol{\beta}}(N+1)})$. Under CKA, the process is as follows. We randomly sample a batch of $m$ states and fed them into the two generated policies $\pi_{\boldsymbol{\theta}=h_{\boldsymbol{\beta}}(N)}$ and $\pi_{\boldsymbol{\theta}=h_{\boldsymbol{\beta}}(N+1)}$. Then, we obtain the output of each layer:

$$X_{\text{input}}^N \in \mathbb{R}^{m \times q_{\text{input}}}, \quad X_{\text{hidden}}^N \in \mathbb{R}^{m \times q_{\text{hidden}}}, \quad X_{\text{output}}^N \in \mathbb{R}^{m \times q_{\text{output}}} \quad (13)$$

for $\pi_{\boldsymbol{\theta}=h_{\boldsymbol{\beta}}(N)}$ and $X_{\text{input}}^{N+1}, X_{\text{hidden}}^{N+1}$, and $X_{\text{output}}^{N+1}$ for $\pi_{\boldsymbol{\theta}=h_{\boldsymbol{\beta}}(N+1)}$, where $q_{\text{input}}, q_{\text{hidden}}$, and $q_{\text{output}}$ are respectively the number of neurons of input, hidden, and output layers of the policies. Now, we can measure the similarity of each layer between the two generated policies, i.e., we have $\rho_{\text{input}}(N) = \text{CKA}(X_{\text{input}}^N, X_{\text{input}}^{N+1})$, $\rho_{\text{hidden}}(N) = \text{CKA}(X_{\text{hidden}}^N, X_{\text{hidden}}^{N+1})$, and $\rho_{\text{output}}(N) = \text{CKA}(X_{\text{output}}^N, X_{\text{output}}^{N+1})$. Details on the computation of the CKA can be found in (Kornblith et al., 2019).

# D    MORE EXPERIMENTAL RESULTS

In this section, we provide more results and analysis to deepen the understanding of our approach.

## D.1    MORE EXPLANATIONS ON THE EXPERIMENTAL RESULTS

From Fig. 3, Panel A, we can see that AugPPO and HyperPPO, as the two most natural methods, are competitive in terms of performance, but both are weaker than our PAPO. We hypothesize that the augmentation or hypernetwork alone would be individually insufficient. (1) In HyperPPO, the population size information needs to first pass through the hypernetwork (which is typically much larger) before passing to the underlying policy network. This could be inefficient when the gradient of the embedding of the population size backpropagates through the deeper hypernetwork, which is similar to the observation in (Sarafian et al., 2021) where the context gradient did not backpropagate through the hypernetwork. (2) In AugPPO, the population size information is directly augmented to the input of the policy network. However, the policy network is less expressive than the hypernetwork as it is typically much smaller than the hypernetwork. Therefore, by inheriting the merits of the two special cases, PAPO could achieve a better performance. However, as mentioned in Sec. 4.2, instead of thoroughly investigating the two special cases and answering the question of which one is better (which could be more involved and outside the scope of our work), we propose a unified framework which encompasses the two options as two special cases.

Given the same training budget as PAPO, PPO-Large and AugPPO-Large cannot always generate approximate Nash policies for games with different population sizes. In this sense, it could be the case that AugPPO-Large could perform worse than PPO-Large as it is more sensitive to the population size. However, we note that we cannot derive the conclusion that AugPPO-Large performs definitely worse than PPO-Large (in "Crowd in Circle" environment, AugPPO-Large performs better than PPO-Large in small-scale settings). Further investigating the difference between PPO-Large and AugPPO-Large could be outside the scope of this work.

## D.2    TRAINING CURVES

In Fig. 9, we present the training curves of different methods in different environments. From the results, we can see that all methods have converged after a sufficient number of training episodes $(2 \cdot 10^7)$. Notice that theoretically computing the exact Nash policy is typically difficult in multi-player games, if not impossible. In this sense, we would expect the methods to return an approximate Nash policy, given that they are trained with a large enough training budget.

During evaluation in a given game, we train a new PPO as the representative agent's BR policy while other agents are fixed to execute the approximate Nash policy. From the representative agent's perspective, the environment is stationary and hence, the problem of learning the BR policy is reduced to an RL problem. In Fig. 10, we present the BR training curves of different methods in a given game. As we can see, the BR training curves have approximately converged, ensuring that the computed approximate NashConv is a reasonable metric to assess the quality of learning.

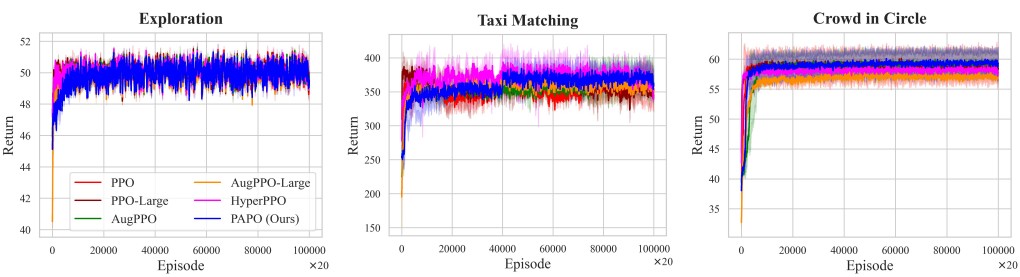

Figure 9: Training curves of different methods in different environments.

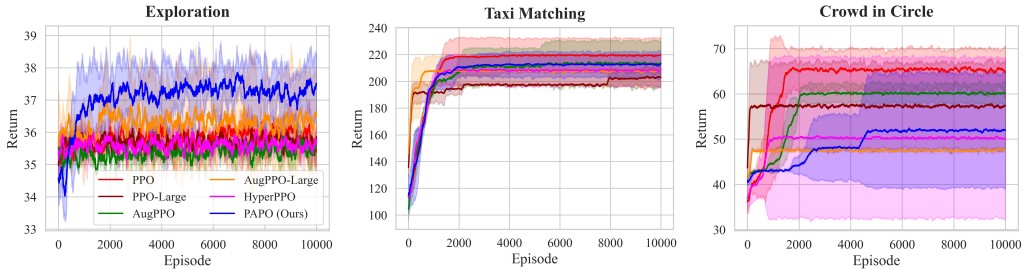

Figure 10: BR training curves of different methods in different environments ($N = 10$).

## D.3   CKA ANALYSIS

In this section, we provide a more detailed analysis on the CKA values. In Fig. 3 (B), we consider the function $\rho(N)$ with the form of $\rho(N) = 1 - \frac{a}{N^b}$. In fact, as mentioned in Sec. 5.1, there could be different forms of $\rho(N)$. However, thoroughly investigating all forms of $\rho(N)$ could be outside the scope of this work. Instead, we choose the one with the form of an inverse polynomial because it coincides with the intuition that when $N$ goes to infinity, the similarity between two policies goes to 1 (i.e., upper-bounded by 1).

Intuitively, the most general case is $\rho(N) = 1 - \frac{a}{cN^b+d}$. However, by computing the p-values, we found that some parameters ($c$ and $d$) are less significant and hence, can be removed from $\rho(N)$. In Fig. 11, we show the curves and in Table 4, we present the values of the parameters and their p-values. We can see that, in most cases, only the parameter $b$ has a significant influence on the curve fitting, i.e., has a small p-value. As a result, in Fig. 3 (B), we consider that $\rho(N) = 1 - \frac{a}{N^b}$ and from the results we can see that the two parameters ($a$ and $b$) are statistically significant, which means that this simpler formula is sufficient to quantitatively characterize the evolution of the CKA values over the population size.

Though the results in Fig. 3 (B) and Fig. 11 provide some intuitions about the evolution of the agents' optimal policies with the population size, they do not rigorously prove the convergence of finite-agent games to MFG. In fact, rigorously proving the convergence could be more involved and maybe only achievable for some special cases such as in (Guo & Xu, 2019). Nevertheless, our results provide an empirical verification, which has not been done before when the policies are represented by DNNs.

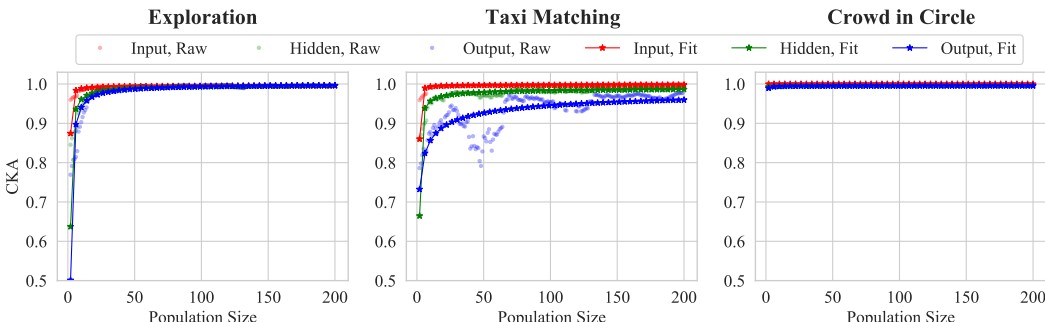

Figure 11: CKA values versus population size. $\rho(N) = 1 - \frac{a}{cN^b+d}$.

Table 4: Values of parameters in $\rho(N) = 1 - \frac{a}{cN^b+d}$ in different game environments.

| Exploration | | | | |
|---|---|---|---|---|
| Layer | $a(p)$ | $b(p)$ | $c(p)$ | $d(p)$ |
| $\rho_{\text{input}}$ | $3.81e^{-6}$ $(0.97)$ | $3.92e^{-5}$ $(0.97)$ | $4.68e^{+0}$ $(\underline{0.00})$ | $-4.68e^{+0}$ $(\underline{0.00})$ |
| $\rho_{\text{hidden}}$ | $4.09e^{-1}$ $(0.99)$ | $6.74e^{-1}$ $(\underline{0.00})$ | $2.98e^{+0}$ $(0.99)$ | $-3.63e^{+0}$ $(0.99)$ |
| $\rho_{\text{output}}$ | $4.22e^{+0}$ $(0.99)$ | $8.93e^{-1}$ $(\underline{0.00})$ | $1.02e^{+1}$ $(0.99)$ | $-1.09e^{+1}$ $(0.99)$ |

| Taxi Matching | | | | |
|---|---|---|---|---|
| Layer | $a(p)$ | $b(p)$ | $c(p)$ | $d(p)$ |
| $\rho_{\text{input}}$ | $8.96e^{-3}$ $(0.99)$ | $1.11e^{+1}$ $(\underline{0.09})$ | $5.71e^{+0}$ $(0.99)$ | $-6.09e^{+0}$ $(0.99)$ |
| $\rho_{\text{hidden}}$ | $2.90e^{-2}$ $(0.99)$ | $1.28e^{-1}$ $(\underline{0.16})$ | $2.32e^{+0}$ $(0.99)$ | $-2.44e^{+0}$ $(0.99)$ |
| $\rho_{\text{output}}$ | $8.43e^{-1}$ $(0.99)$ | $4.52e^{-1}$ $(\underline{0.02})$ | $1.84e^{+0}$ $(0.99)$ | $+6.26e^{-1}$ $(0.99)$ |

| Crowd in Circle | | | | |
|---|---|---|---|---|
| Layer | $a(p)$ | $b(p)$ | $c(p)$ | $d(p)$ |
| $\rho_{\text{input}}$ | $-2.95e^{-4}$ $(0.91)$ | $-7.77e^{-1}$ $(\underline{0.16})$ | $1.75e^{+0}$ $(0.91)$ | $-1.62e^{+0}$ $(0.91)$ |
| $\rho_{\text{hidden}}$ | $+1.49e^{-4}$ $(0.99)$ | $+2.65e^{-4}$ $(0.99)$ | $1.46e^{+1}$ $(0.99)$ | $-1.45e^{+1}$ $(0.99)$ |
| $\rho_{\text{output}}$ | $-5.13e^{-2}$ $(0.99)$ | $-3.43e^{-1}$ $(\underline{0.47})$ | $1.05e^{+1}$ $(0.99)$ | $-1.32e^{+1}$ $(0.99)$ |

## D.4 REPRESENTATIONS OF GAME STATES

In this section, we provide more results of the representations of different game states. To this end, we first manually construct some natural and understandable game states according to the position of an agent on the map/circle. For example, the game state "Bottom-left" means the agent is located on the bottom-left of the map at any time. In Fig. 12 – Fig. 14, we show the UMAP embeddings of different game states. We found that the UMAP embeddings of some game states (e.g., bottom-left and upper-right in Exploration, bottom-left and bottom-right in Taxi Matching, center in Crowd in Circle) present a similar trend as mentioned in the main text while the UMAP embeddings of other game states do not have clear patterns. We hypothesize that some manually constructed high-level (coarse-grained) game states may consist of very different underlying (fine-grained) states of the agents. On the other hand, to some extent, the results imply that our PAPO may not be a smooth enough function of the population size. In this sense, in future works, proposing novel techniques to drift PAPO toward a smoother function of the population size is worth interesting and may further improve the performance of PAPO, making it a more elegant solution.

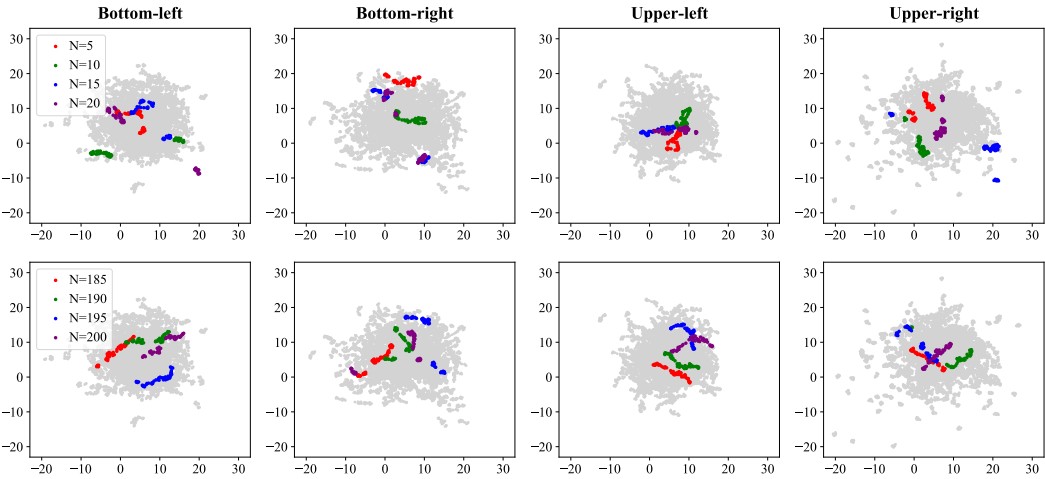

Figure 12: UMAP embeddings of different game states in Exploration.

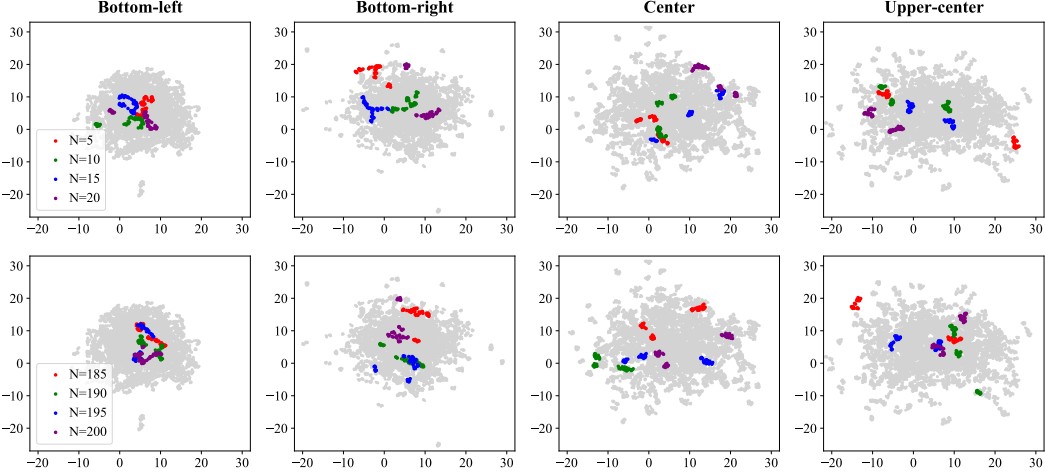

Figure 13: UMAP embeddings of different game states in Taxi Matching.

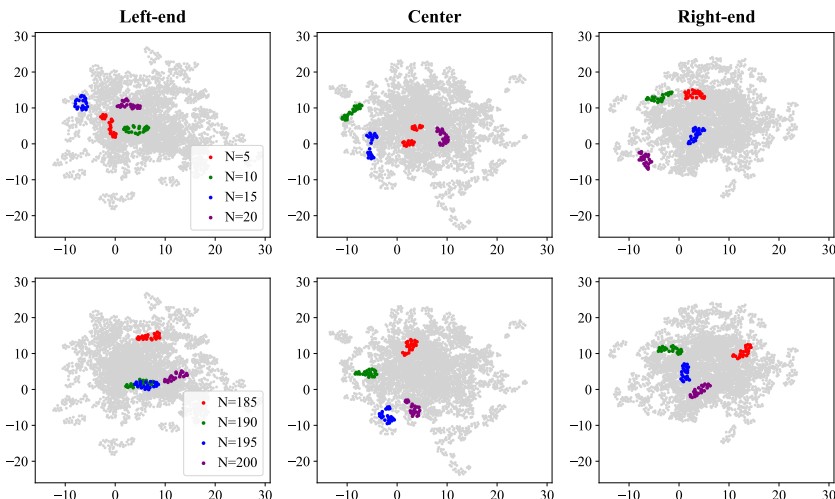

Figure 14: UMAP embeddings of different game states in Crowd in Circle.

### D.5   Effect of Population-size Encoding

In this section, we provide more analysis to explore the effect of population-size encoding on the training process. In Fig. 4, we can see that the training of PAPO is collapsed when using RE. Though the approximate NashConv is less meaningful in this case, for completeness, we present the resulting approximate NashConv in Fig. 15–Fig. 16. The results in Fig. 15 correspond to Fig. 1 which considers small-scale settings in Taxi Matching environment. The results in Fig. 16 correspond to Fig. 3 which considers larger-scale settings in different environments.

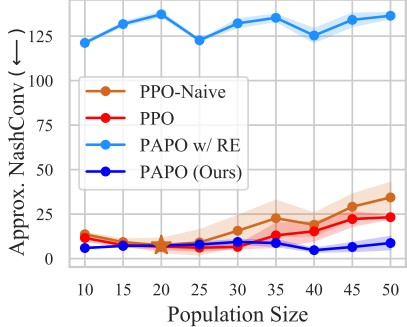

Figure 15: Approximate NashConv versus population size in small-scale settings in Taxi Matching environment.

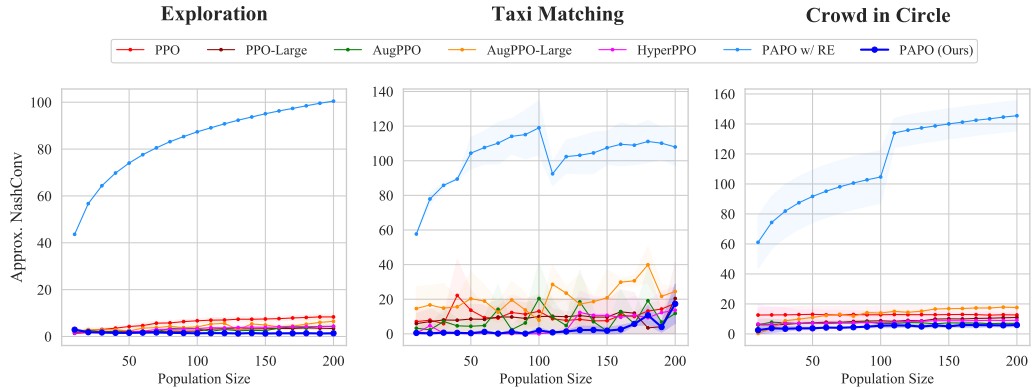

Figure 16: Approximate NashConv versus population size in large-scale settings in different environments.

It is well known that increasing the entropy of a policy at the beginning of training can incentivize it to explore the environment more widely and in turn improve the overall performance of the policy (Haarnoja et al., 2018; Ahmed et al., 2019). In this sense, given the initialized PAPO, then we expect that the generated policies have high entropy, regardless of the input population size. More precisely, given any $N$, the generated policy $\pi_{\boldsymbol{\theta}=h_{\boldsymbol{\beta}}(N)}$ by the initialized PAPO would output an approximate uniform distribution over the action space $\mathcal{A}$ for a given state $s_t$. Let $\hat{\pi}$ be the uniform policy, i.e., $\hat{\pi}(a|s_t) = \frac{1}{|\mathcal{A}|}$ for all $a \in \mathcal{A}$. We can compute the KL-divergence between $\pi_{\boldsymbol{\theta}=h_{\boldsymbol{\beta}}(N)}$ and $\hat{\pi}$, denoted as $\kappa(N) = \mathrm{D_{KL}}\big(\pi_{\boldsymbol{\theta}=h_{\boldsymbol{\beta}}(N)}\|\hat{\pi}\big)$. Then, intuitively, at the beginning of training, $\kappa(N)$ is near 0 for different $N$'s. After the neural networks of PAPO are well trained, $\kappa(N)$ will be larger than 0 and vary with $N$.

In Fig. 17, we present the results to verify the aforementioned conclusions. From the results, we can see that, when using RE, the approximate uniform action distribution only holds for small $N$'s, as shown in the first row. However, $\kappa(N)$ quickly increases when $N$ is increasing, which could severely hamper the training of PAPO (see the comparison between "PAPO w/ RE (initial)" and "PAPO w/ RE (trained)"). In striking contrast, using BE can effectively avoid such a training collapse.

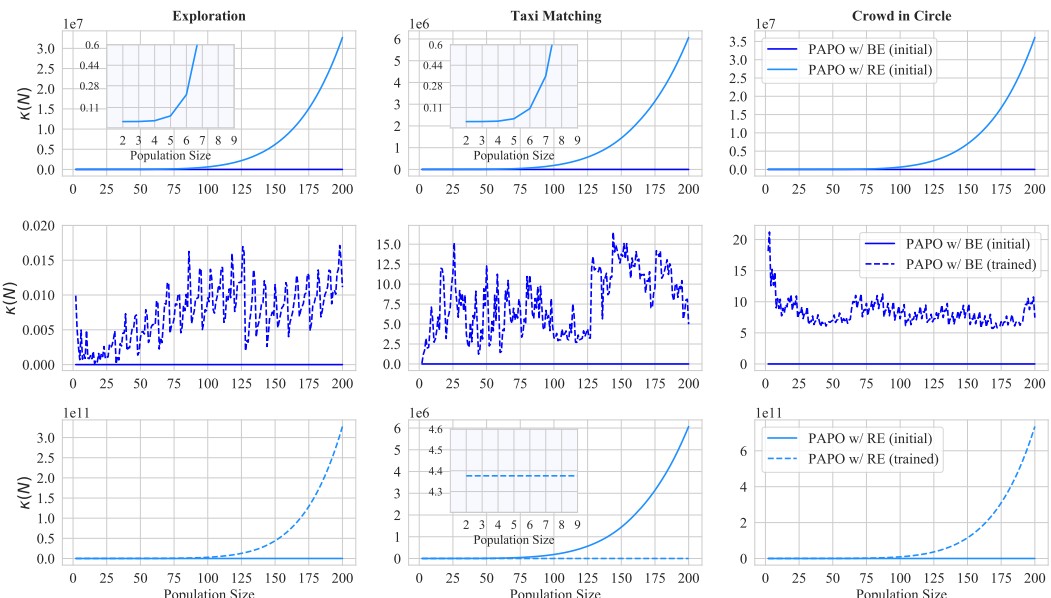

Figure 17: The KL divergence $\kappa$ versus population size.

## D.6 REWARD DISTRIBUTION SHIFT

One of the reasons we hypothesize why PPO performs poorly across different games is the reward distribution shift in the environments. To verify the intuition, for an environment, we use a randomly initialized policy to sample 1,000 trajectories (episodes) and get the reward distribution by statistics. As shown in Fig. 18–Fig. 20, the reward distribution varies sharply with the increasing population size. Intuitively, PPO works poorly across the games as it does not consider the increasing diversity of reward signals resulting from the changes in population size. In contrast, PAPO (and other population-size-aware methods) possesses the ability to work consistently well across different games by explicitly taking the information of the population size into account during training.

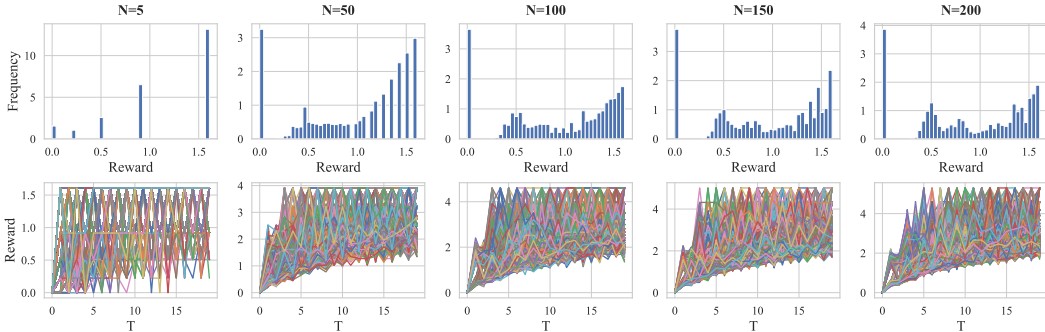

Figure 18: Sampled reward distributions and trajectories in Exploration.

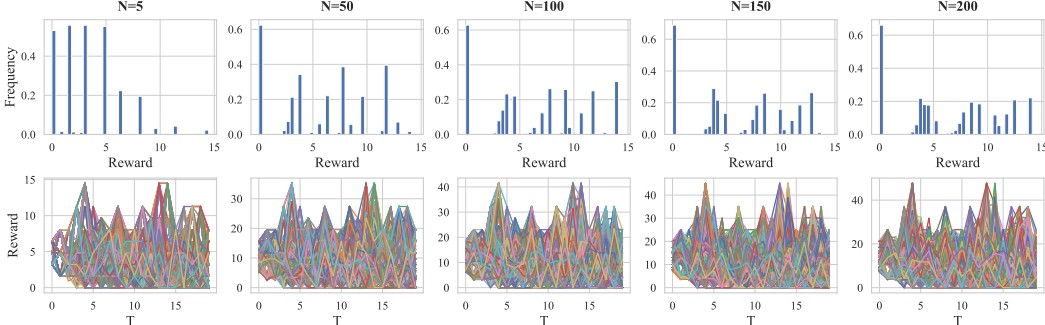

Figure 19: Sampled reward distributions and trajectories in Taxi Matching.

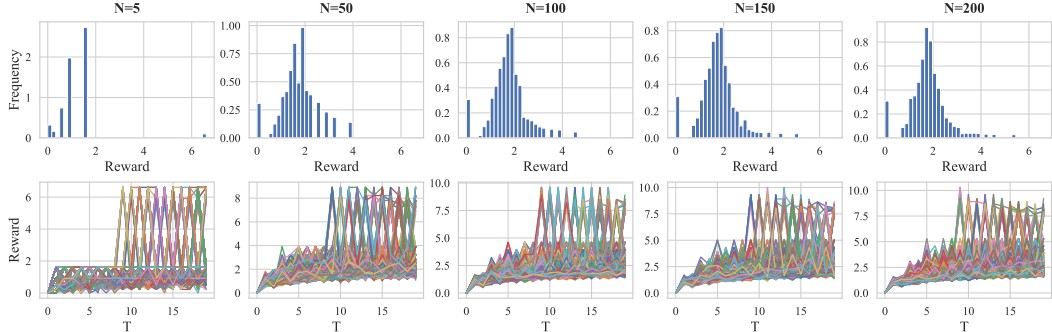

Figure 20: Sampled reward distributions and trajectories in Crowd in Circle.

