# OpenReview forum: "Population-size-Aware Policy Optimization for Mean-Field Games"
_ICLR.cc/2023/Conference — ICLR 2023 poster_

### Official Review · Reviewer_BBrB · 2022-10-24

**Confidence:** 4
**Correctness:** 4
**Technical Novelty And Significance:** 4
**Empirical Novelty And Significance:** 4
**Recommendation:** 6

**Clarity, Quality, Novelty And Reproducibility:**

Typo:
Page 26: “posses” → “possesses”


**Strength And Weaknesses:**

Strength: From my point of view, the paper is well written and the question is interesting for the mean field game community. The fact that several examples from the literature are treated is also a positive point.

Weaknesses:
(1) It is not clear to me why the policies are stationary although the time horizon is finite. In principle, time-dependent policies could achieve a better reward than stationary ones.

(2) I find it difficult to really understand the baselines. Since this is a numerical paper, I think it would be important to provide even more details in Appendix C.2. For example, the first method is simply referred as standard PPO but it is not really clear what the inputs are. Is it only the agent’s state (and the current time)?


**Summary Of The Paper:**

The main topic of this paper is mean field games, which have been developed to study games with a very large number of players. A mean field game correspond to the asymptotic limit where the number of players becomes infinite. Here, the authors propose to combine the mean field point of view with a hypernetwork to learn policies that can be adapted to different (finite) sizes of populations. After introducing the machine learning model, they propose an algorithm to train the network. Last, the paper presents numerical experiments on several examples, by showing how the policy learnt by the algorithm provides and approximate Nash equilibrium on games of various population sizes. The authors also provide some measure of how the learned policy changes with the population size.

**Summary Of The Review:**

To the best of my knowledge, this paper proposes a new approach to use learn a policy network that can be efficient for various population sizes. The extra flexibility is probably interesting for applications. I would recommend accepting the paper, although I believe that addressing the above questions would strengthen the contributions.

---

> ### Author Response · Authors · 2022-11-18
> **Response to Reviewer BBrB**
>
> Thanks for your constructive suggestions, and we are glad that you found that the question is interesting for the mean-field game community. We summarize the questions and present our responses below.
>
> ---
> **Q1. It is not clear to me why the policies are stationary although the time horizon is finite. In principle, time-dependent policies could achieve a better reward than stationary ones.**
> * In the revision, we redefine the policy such that it is a mapping from the cartesian product of the state space and the set of time indices to the action space. In experiments, we follow [1] to make the policy dependent on time by concatenating the state with time (footnote 2).
> ---
> **Q2. I find it difficult to really understand the baselines. Since this is a numerical paper, I think it would be important to provide even more details in Appendix C.2. For example, the first method is simply referred as standard PPO but it is not really clear what the inputs are. Is it only the agent’s state (and the current time)?**
> * We follow the reviewer's suggestion to provide more details in Appendix C.2 in the revision. We use Figure 8 to intuitively present the algorithm frameworks of different methods and also provide detailed descriptions. For PPO/PPO-Large, the input is the concatenation of the agent's state and the current time (see the revision for the details of other methods).
> * In addition, in Appendix B.1, we provide the structure of the hypernetwork (Figure 6) and the code snippet (Figure 7) in the forward process of PAPO, which further improves the reproducibility of our work (also as a response to the Q.4 of the reviewer cUv7).
> ---
> **Q3. Typo: Page 26: "posses" $\to$ "possesses".**
> * We thank the reviewer for pointing out the typo. In the revision, we fix this and other typos.
> ---
> [1] Mathieu Lauri\`ere et al. Scalable deep reinforcement learning algorithms for mean field games. In ICML, pp. 12078–12095, 2022.

---

### Official Review · Reviewer_g5d3 · 2022-10-26

**Confidence:** 3
**Correctness:** 3
**Technical Novelty And Significance:** 3
**Empirical Novelty And Significance:** 4
**Recommendation:** 6

**Clarity, Quality, Novelty And Reproducibility:**

Overall the clarity is good but there are some issues with the paper that I'll list below:

1) I feel like the title isn't suited to the paper. When I saw "scaling laws" in the title I thought it referred to the size of the neural network (as is typical in machine learning papers these days). The authors apparently intend it to mean the number of agents, but even in that case it doesn't really make sense. This paper isn't about how performance and behavior changes as the number of agents grows. It's about finding a way to generate a policy that generalizes across arbitrary numbers of agents.

2) The authors reference Figure 1 in the intro but don't sufficiently explain it until the results section of the paper. I'm left wondering what this environment is, and what you are "transferring" from and to.

3) The number of citations in this paper is... excessive. The related works section is basically a list of any paper that might be slightly related. This is space that could be better used explaining this paper or explaining how a select set of papers are actually related.

4) It wasn't clear to me until the related work section that this paper is limited to Markov Games. I think that should be made clear in the abstract or at least the intro.

5) Saying Raw Encoding "cannot work in practice" seems like a strong statement that would require more explanation if you're going to make that claim. Is there simply a lack of evidence that it works, or is there some reason to think that it definitely cannot work?

6) It would be nice to see more explanation of why PAPO benefits from both augmentation and a hypernetwork. Is there a reason to think there might be some synergy between the two or are their contributions largely independent? Is there a reason to think that augmentation or hypernetwork alone would be individually insufficient in larger-scale experiments?

7) There is a typo in the last sentence of section 4.

8) In the figures, PPO-Large and AugPPO-Large have almost the same color, which makes them difficult to distinguish. However, if I'm reading the figure correctly, it looks like AugPPO-Large is doing universally worse than PPO-Large. Is there any reason to think this should be the case? Is this mislabeled?

**Strength And Weaknesses:**

Strengths: The success of binary encoding seems like the highlight of the paper to me. The fact that it works across all the games and does so much better than raw encoding is non-obvious and useful. The experiments and ablations are also well done and the paper was well-written.

Weaknesses: I think the paper uses inappropriate baselines. Figure 1, for example, compares PAPO to two baselines. The first is PPO-naive which shows how PPO trained on a game variant with 20 agents generalizes to game variants with 2 <= N <= 50 agents. The second is PPO, which shows how PPO trained on 2 <= N <= 50 agents (but without N as an input to the network) performs in game variants with 2 <= N <= 50 agents. Both of these baselines will clearly perform poorly because they don't condition on the actual game state. In my opinion the appropriate baseline would be the Raw Encoding option.

**Summary Of The Paper:**

This paper investigates the problem of training neural network policies in many-player games that are capable of generalizing to any number of player, rather than just a specific number N. They accomplish this through two methods. The first is augmentation, in which the number of players N is an input into the policy network. The second is a hypernetwork that takes N as input and outputs the parameters of the neural network to use for the game that has N agents. They show in experiments that this beats baseline techniques such as PPO that does not take the number of players as input.

**Summary Of The Review:**

I think there is a core contribution to this paper that is valuable and validated well in experiments, but I think the framing of the paper, particularly regarding the baselines, would make it much stronger.

---

> ### Author Response · Authors · 2022-11-18
> **Response to Reviewer g5d3**
>
> Thanks for your valued feedback and your appreciation that there is a core contribution to this paper that is valuable and validated well in experiments. We summarize the questions and present our responses below.
>
> ---
> **Q1. I think the paper uses inappropriate baselines. ... In my opinion the appropriate baseline would be the Raw Encoding option.**
> * We agree with the reviewer that the methods using RE (Raw Encoding) can be the baselines. However, they could be weaker than those using BE due to the possible training collapse. Therefore, we use the methods using BE as the baselines, which ensures a fair comparison between them and our PAPO (as well as the fair comparison between these baselines), while investigating the effect of RE on the performance in the ablation study. In the revision, we include this discussion in Appendix C.2. Nevertheless, we show the evaluation performance (approximate NashConv) of PAPO with RE in Appendix D.4 for completeness.
> ---
> **Q2. I feel like the title isn't suited to the paper. When I saw "scaling laws" in the title I thought it referred to the size of the neural network (as is typical in machine learning papers these days). The authors apparently intend it to mean the number of agents, but even in that case it doesn't really make sense. This paper isn't about how performance and behavior changes as the number of agents grows. It's about finding a way to generate a policy that generalizes across arbitrary numbers of agents.**
> * For motivation, we attempt to bridge the two fields of finite-agent MGs and MFGs in the context of DNN-based policies. Specifically, we aim to investigate how the optimal policies of agents evolve with the population size (as the reviewer said, how the behavior of the agents' policies changes as the number of agents grows). This is the first question which is discussed in Introduction and Problem Statement.
> * The premise to investigate the scaling laws is that the agents' policies are indeed the (approximate) Nash policies. However, there are several challenges including the theoretical intractability, computational difficulty, and sub-optimality (as discussed in Introduction and Problem Statement). Therefore, we need to propose a novel approach to efficiently generate efficient policies for a set of finite-agent games (as the reviewer pointed out, finding a way to generate a policy that generalizes across arbitrary numbers of agents) before we can further investigate the scaling laws. This is the second question which is discussed in Introduction and Problem Statement.
> * In the above sense, we think that the title is suited to this work because the two questions are closely related, though we need to first propose a novel approach to efficiently generate efficient policies before further investigating the scaling laws. Nevertheless, we are welcome for more discussion if the reviewer has any more suggestions.
> ---
> **Q3. The authors reference Figure 1 in the intro but don't sufficiently explain it until the results section of the paper. I'm left wondering what this environment is, and what you are "transferring" from and to.**
> * In Introduction, we explicitly point out the environment used in Figure 1. Besides, we restate the PPO-Naive method such that its operation is easier to understand: directly apply the policy trained in a given population size ($N=20$) to other population sizes.
> ---
> **Q4. The number of citations in this paper is... excessive. The related works section is basically a list of any paper that might be slightly related. This is space that could be better used explaining this paper or explaining how a select set of papers are actually related.**
> * In the revision, we follow the reviewer's suggestion to reorganize the Related Work section. We discuss the most related works in Section 2, which more accurately positions our work within the literature to avoid the misunderstanding that our contribution is over-claimed. Other related works have been deferred to Appendix A.3.
> ---
> **Q5. It wasn't clear to me until the related work section that this paper is limited to Markov Games. I think that should be made clear in the abstract or at least the intro.**
> * In the revision, in Introduction, we follow the reviewer's suggestion to emphasize that our work considers the Markov games which have a similar structure to the MFG (footnote 1).

---

> > ### Author Response · Authors · 2022-11-18
> > **Response to Reviewer g5d3 (continued)**
> >
> > **Q6. Saying Raw Encoding "cannot work in practice" seems like a strong statement that would require more explanation if you're going to make that claim. Is there simply a lack of evidence that it works, or is there some reason to think that it definitely cannot work?**
> > * In Section 4.1, the first paragraph, we provide some explanations for not choosing the RE. But as the training collapse is only an experimental observation in our work, we agree with the reviewer that "cannot work in practice" is a strong statement. Therefore, in the revision, we weaken this claim by saying that "This might work poorly in practice", which is more consistent with the experimental observations in this work.
> > ---
> > **Q7. It would be nice to see more explanation of why PAPO benefits from both augmentation and a hypernetwork. Is there a reason to think there might be some synergy between the two or are their contributions largely independent? Is there a reason to think that augmentation or hypernetwork alone would be individually insufficient in larger-scale experiments?**
> > * We hypothesize that the augmentation or hypernetwork alone would be individually insufficient. (1) In HyperPPO, the population size information needs to first pass through the hypernetwork (which is typically much larger) before passing to the underlying policy network. This could be inefficient when the gradient of the embedding of the population size backpropagates through the deeper hypernetwork, which is similar to the observation in [1] where the context gradient did not backpropagate through the hypernetwork. (2) In AugPPO, the population size information is directly augmented to the input of the policy network. However, the policy network is less expressive than the hypernetwork as it is typically much smaller than the hypernetwork. Therefore, by inheriting the merits of the two special cases, our PAPO could achieve better performance. However, as mentioned in Section 4.2, instead of thoroughly investigating the two special cases and answering the question of which one is better (which could be more involved and outside the scope of our work), we propose a unified framework which encompasses the two options as two special cases.
> > * Following the reviewer's suggestion, we include the above explanations in the revision, see Appendix D.1.
> > ---
> > **Q8. There is a typo in the last sentence of section 4.**
> > * Thanks for pointing out this typo. In the revision, we fix this and other typos.
> > ---
> > **Q9. In the figures, PPO-Large and AugPPO-Large have almost the same color, which makes them difficult to distinguish. However, if I'm reading the figure correctly, it looks like AugPPO-Large is doing universally worse than PPO-Large. Is there any reason to think this should be the case? Is this mislabeled?**
> > * In the revision, we change the line colors of PPO-Large and AugPPO-Large to make them easier to distinguish.
> > * We have checked the results carefully and found that they are not mislabeled. As shown in experiments, given the same training budget as PAPO, PPO-Large and AugPPO-Large cannot always generate approximate Nash policies for games with different population sizes. In this sense, it could be the case that AugPPO-Large could perform worse than PPO-Large (in most cases or as said by the reviewer that AugPPO-Large is doing universally worse than PPO-Large) as it is more sensitive to the population size. However, we note that we cannot derive the conclusion that AugPPO-Large performs definitely worse than PPO-Large (in "Crowd in Circle", AugPPO-Large performs better than PPO-Large in small-scale settings), given that both of them cannot always generate approximate Nash policies.
> > * In the revision, we include the above discussion in Appendix D.1. But with a gentle reminder, further investigating the difference between PPO-Large and AugPPO-Large could be outside the scope of this work.
> > ---
> > [1] Elad Sarafian et al. Recomposing the reinforcement learning building blocks with hypernetworks. In ICML, pp. 9301–9312, 2021.

---

> > > ### Comment · Reviewer_g5d3 · 2022-11-25
> > > **Regarding "Scaling Laws"**
> > >
> > > I'm still not convinced that "Scaling Laws" is the right term to use for the title of this paper. I've only seen it being used in the context of scaling things like the model size. Could the authors point to similar works that use the term "scaling laws" in a way that's closer to how it's being used in this paper?

---

> > > > ### Author Response · Authors · 2022-11-26
> > > > **Response to the concern about the term "Scaling Laws"**
> > > >
> > > > Dear Reviewer g5d3,
> > > >
> > > > Thanks for your further comments. For your concerns about the term "Scaling Laws", we provide more explanations as follows.
> > > >
> > > > * The term "Scaling Law" has been widely used in different areas including biology, physics, social science, and computer science. It typically describes the functional relationship between two quantities. In this sense, we note that the two quantities are typically problem-dependent, e.g., the performance of a model and the model size as mentioned by the reviewer, the fluctuations in the number of messages sent by members in a communication network and their level of activity [1], the probability that a vertex in a social network interacts with $k$ other vertices and the $k$ other vertices [2], to name a few.
> > > >
> > > > * In our context, the two quantities are: (1) the behavior of the policy network, and (2) the number of agents. In fact, this is conceptually similar to the scaling laws in social networks which investigate how does some quantity (e.g., the property of a social network such as the connectivity) change with the number of vertices (typically, a vertex stands for an individual, i.e., an agent) [2]. So, our work also follows a similar idea but focuses on investigating how does the behavior of the policy network change with the number of agents. Therefore, we feel that the term "Scaling Law" is suited to our work as, in a more general sense, it can be used to describe the relationship between any two quantities which are determined by the specific problem at hand, not only the size of model or training set or the amount of compute used for training. Furthermore, in contrast to the areas such as NLP, CV, and single-agent RL which typically consider a single model, it is natural to consider the scaling laws of the policy with the number of agents in multi-agent systems (MAS).
> > > >
> > > > If possible, we will add the above explanations in the "final" version to facilitate the understanding of the scaling laws in this paper.
> > > >
> > > > If the above further explanations still cannot convince you, do you have any suggestions about the term? We believe our problem is important, and we are glad to discuss with you for a more accurate term of the problem investigated in this paper.
> > > >
> > > > [1] Diego Rybski et al. Scaling laws of human interaction activity. PNAS, 106(31), 12640-12645, 2009.
> > > >
> > > > [2] Albert-L\'aszl\'o Barab\'asi et al. Emergence of Scaling in Random Networks. Science, 286(5439), 509-512, 1999.

---

### Official Review · Reviewer_BxNp · 2022-11-02

**Confidence:** 2
**Correctness:** 2
**Technical Novelty And Significance:** 1
**Empirical Novelty And Significance:** 3
**Recommendation:** 5

**Clarity, Quality, Novelty And Reproducibility:**

- Some aspects of the problem description can be confusing:
  - It should be stressed that the finite-player Markov Games (MG) considered in this work are limited to those that are finite-player approximations of mean-field games. This property is described in the paper by being "homogeneous", which does not seem to be its standard usage. Note that this restriction is heavy: the MG has to be symmetric, and interaction can only happen through states., and not even the paper-scissors-rock game can fit within the definition of MG of this work. Although this formulation is sufficient enough for the setting of this work, it should nevertheless be emphasized that this work does not solve the standard MG problem.
  - The "scaling law" is limited to the similarity measure of $\pi^\ast_N$ and $\pi^\ast_{N+1}$. This approach (1) is not well defined considering Nash Equilibria may not be unique; (2) may not be a good measure at all of whether the Nash equilibria at time $N$ will perform well for different number of players.
- As far as I can tell, the algorithmic techniques, including the binary encoding of $N$ and the specific way to include hypernetworks, are novel.


**Strength And Weaknesses:**

**Strength:**
1. The problem that is studied, namely Markov games with varying number of players, is interesting and relevant.
2. The experimental results seem fair and well-represented.

**Weaknesses:**
1. The contribution of this paper is largely overstated:
  - The abstract claims that this work is "the first attempt to bridge the two research fields [finite-agent MG and MFG]", overlooking existing work on the theoretical foundation for finite-agent approximation (e.g. Saldi et al.) and algorithmic design (e.g. Li et al.).
  - The title, the abstract and the introductory section promise extensive analysis of the "scaling law", which is revealed to be only a presentation of the similarity measure of output policies for one specific algorithm. There is no analysis of its meaning beyond a simple curve-fitting.
2. The writing of this paper has much to be desired.


-----------
Saldi et al. Markov–Nash equilibria in mean-field games with discounted cost. 2018

Li et al. Permutation Invariant Policy Optimization for Mean-Field Multi-Agent Reinforcement Learning: A Principled Approach. 2021.

**Summary Of The Paper:**

This paper studies the learning of infinite-player Mean-Field Games (MFG) through training on finite-agent Markov Game (MG) approximations. By combining two existing techniques, namely augmentation and hypernetworks, the authors propose a PPO-based algorithm called PAPO, which is demonstrated to achieve better performance in several mean-field game environments.

**Summary Of The Review:**

Although some solid empirical effort is made to deal with a relevant problem, this paper offers little understanding of any "scaling law" and greatly overstates its contribution. I cannot recommend acceptance of this paper in its current form.

---

> ### Author Response · Authors · 2022-11-18
> **Response to Reviewer BxNp**
>
> Thanks for your helpful comments. We summarize the questions and present our responses below.
>
> ---
> **Q1. The contribution of this paper is largely overstated: $\bullet$ The abstract claims that this work is "the first attempt to bridge the two research fields [finite-agent MG and MFG]", overlooking existing work on the theoretical foundation for finite-agent approximation (e.g. Saldi et al.) and algorithmic design (e.g. Li et al.). $\bullet$ The title, the abstract and the introductory section promise extensive analysis of the "scaling law", which is revealed to be only a presentation of the similarity measure of output policies for one specific algorithm. There is no analysis of its meaning beyond a simple curve-fitting.**
> * In the revision, we reorganize the Related Work section to position our work within the literature more accurately. Meanwhile, the Abstract and Introduction are also properly modified accordingly. Given that our work is accurately positioned within the literature, a more accurate claim is: our work is the first attempt to establish the connection between the MFG and the corresponding finite-agent MGs when agents' policies are represented by deep neural networks (DNNs). In this sense, we note that it is a misunderstanding that our contribution is over-claimed (see also the response to Q.3 of the reviewer cUv7).
> * For the scaling laws, we provide more explanations as follows. (1) As our work is the first attempt to investigate the scaling laws in the context of DNN-based policies (in the MFG and the corresponding finite-agent games), we identify the proper measure to capture how the policies evolve with the population size, see "Similarity Measure" in Section 5. (2) In Section 5.1, Paragraph 3, we first show the raw values of similarity and the fitting curves which are the most intuitive results to show the evolution of the generated policies. Then, in Paragraph 4, we provide more explanations on how the results relate to the common knowledge/conclusions in deep learning. Finally, in Paragraph 5 and Appendix D.3, we further study the difference between the generated policies by investigating how the policies encode different human-understandable game states (similar to [1] which investigated how the policy encodes different game states such as "agent in home base"). Therefore, we argue that the analysis of the evolution of the generated policies in our work does not only limit to a simple curve-fitting.
> * Though more analysis of the scaling laws of the policies could be possible (in the revision, we weaken the claim of "extensive analysis of the scaling laws"), we note that an important premise is that the generated policies are indeed approximate Nash policies (i.e., the policies have good performance in terms of approximate NashConv). Therefore, we propose a novel approach to efficiently generate efficient policies for a set of games with different population sizes, which is one of the contributions of our work.
>
> ---
> **Q2. The writing of this paper has much to be desired.**
> * In the revision, we check the writing carefully and make the conclusions more rigorous to avoid the misunderstanding that our contribution is over-claimed. We are welcome for more discussion if the reviewer has any more suggestions.
>
> ---
> **Q3. It should be stressed that the finite-player Markov Games (MG) considered in this work are limited to those that are finite-player approximations of mean-field games.**
> * In this work, we consider the MGs that have a similar structure to the MFG (as the reviewer mentioned that the finite-player approximations of MFGs) (also as discussed in Appendix A.2). In the revision, we follow the reviewer's suggestion to emphasize this setting (footnote 1).
> * Though the game model considered in this work is a subclass in game theory, it has been widely used to model many large-scale multi-agent scenarios in literature such as [2,3] (see also Appendix A.3).

---

> > ### Author Response · Authors · 2022-11-18
> > **Response to Reviewer BxNp (continued)**
> >
> > **Q4. The "scaling law" is limited to the similarity measure of $\pi_{N}^{\star}$ and $\pi_{N+1}^{\star}$. This approach (1) is not well defined considering Nash Equilibria may not be unique; (2) may not be a good measure at all of whether the Nash equilibria at time $N$ will perform well for different number of players.**
> > * We note that $\rho(\pi_N^*, \pi_{N+1}^*)$ (to be more accurate, in the revision, we refer to this as the difference) can be any type of measure (as pointed out in the last paragraph in Section 4.2). In our work, as the agents' policies are represented by DNNs, the most straightforward and natural idea is to measure the similarity between the DNNs.
> > * Though there could be multiple Nash equilibria, computing them is typically intractable in complex multi-agent games. Therefore, the common practice is to train an approximate Nash equilibrium (note that for a given trained policy, one can verify whether it is an approximate Nash policy by computing the approximate NashConv, which is the common practice in the literature). In this sense, the similarity between DNNs is a reasonable measure to capture the evolution of the policies with the population size.
> > * The critical premise to investigate the scaling laws is that the agents' policies are indeed the (approximate) Nash policies. This corresponds to the question mentioned by the reviewer: whether the Nash equilibria at population size (note that not time) $N$ will perform well for different numbers of players (in fact, this is the PPO-Naive introduced in Introduction: directly apply the Nash policy at a given game to other games with different numbers of agents). So, before investigating the scaling laws, we first need to propose novel approaches to efficiently generate efficient policies (i.e., the policies that have low NashConv, as mentioned in footnote 3) for games with different population sizes, which is one of the contributions of our work.
> >
> > ---
> > [1] Max Jaderberg et al. Human-level performance in 3D multiplayer games with population-based reinforcement learning. Science, 364(6443):859–865, 2019.
> >
> > [2] Duc Thien Nguyen et al. Credit assignment for collective multiagent RL with global rewards. NeurIPS, pp. 8102–8113, 2018.
> >
> > [3] Xin Guo et al. Learning mean-field games. NeurIPS, pp.
> > 4966–4976, 2019.

---

> ### Author Response · Authors · 2022-11-25
> **Further Discussions**
>
> Dear Reviewer BxNp,
>
> Thanks for your valuable comments which help the improvement of our work.
>
> We would appreciate it if you could let us know whether our responses and revision have addressed your concerns or not. We would be happy to do any follow-up discussion if you have any other questions or suggestions.
>
> Sincerely,
>
> Authors of Paper3814

---

> ### Author Response · Authors · 2022-12-01
> **Looking forward to your feedback**
>
> Dear Reviewer BxNp,
>
> Thank you again for your time and effort in reviewing our work and your valuable comments which help improve our draft.
>
> We have posted our responses to your initial comments and uploaded the revised draft, which we believe has covered your concerns. We would like to give a summary of our responses and revision: (1) To avoid the misunderstanding that the contribution of this work is over-claimed, in the revision, we reorganize the Related Work section to more accurately position our work within the literature. (2) We provide more explanations to illustrate the similarity measure used in this work and the scaling laws investigated in this work. (3) We follow your suggestions to make the conclusions more rigorous in the revision. For details, please check our responses and revised draft where the major changes have been highlighted in blue.
>
> Since the end of discussion stage 2 is approaching, this is a gentle reminder that whether our responses and the revision have properly addressed your concerns. We are looking forward to your reply to our responses, and we are open to discussions if you have any further comments or suggestions.
>
> Sincerely,
>
> Authors of Paper3814

---

### Official Review · Reviewer_cUv7 · 2022-11-04

**Confidence:** 4
**Correctness:** 3
**Technical Novelty And Significance:** 3
**Empirical Novelty And Significance:** 3
**Recommendation:** 8

**Clarity, Quality, Novelty And Reproducibility:**

The writing is clear and relatively easy to follow. The generalization across different games with different number of players is novel, but I think the authors are over-claiming their contribution in terms of studying the scaling laws. There have been numerous works studying the convergence behavior of policies from both the theoretical and computational viewpoints as the number of players goes to infinity, like [A,B] and some other papers citing [C] (a few with rates/scaling laws of convergence as well), to name just a few. But the scaling law study in this paper seems to be more general than existing works, despite the approximation/inaccuracies mentioned above. The reproducibility looks good in general, but the detailed hyper-network structure in Appendix B.1 might better be made clearer with an illustration figure, and the shapes of $x$, $w_2(z)$, $g_2(z)$ and $b_2(z)$ and why they match might better be made clearer as well.

[A] Guo, X., & Xu, R. (2019). Stochastic games for fuel follower problem: N versus mean field game. SIAM Journal on Control and Optimization, 57(1), 659-692.

[B] Cabannes, T., Lauriere, M., Perolat, J., Marinier, R., Girgin, S., Perrin, S., ... & Elie, R. (2021). Solving N-player dynamic routing games with congestion: a mean field approach. arXiv preprint arXiv:2110.11943.

[C] Saldi, N., Basar, T., & Raginsky, M. (2018). Markov--Nash equilibria in mean-field games with discounted cost. SIAM Journal on Control and Optimization, 56(6), 4256-4287.

**Strength And Weaknesses:**

Strengths:
* This paper proposes a way to learn policies that generalize across games with different number of players. To my knowledge, this is new.
* This paper studies the scaling laws of policies as the number of players goes to infinity.

Weaknesses:
* Too much approximation/inaccuracies (both in terms of the algorithms used to compute the policies and the methods used to evaluate the policies) in the study of the scaling laws, making it not very convincing.
* The convergence result of independent policy gradient in Appendix A does not seem to be right. The authors are confusing symmetric games with common interest games, and Proposition A.1.1 is incorrect. In fact, if Proposition A.1.1 is correct, then one should expect that all mean-field games are potential, which is wrong (and there were other earlier papers making such similar common mistakes). Also that's why I say that there are too much approximation, since there is indeed no clue of whether independent RL converges for symmetric (but not potential/common interest) games, even in the simplest settings (e.g., without function approximation, etc.). But this is not a major result of the paper so I think it's a relatively minor weakness. But please double check carefully and fix this issue if needed in any case.

**Summary Of The Paper:**

This paper studies the scaling laws of mean-field games by proposing a variant of independent proximal policy optimization (PPO), called Population-size-Aware Policy Optimization (PAPO), that trains Nash equilibrium agent policies that generalize across different number of agents in homogeneous games by utilizing hyper-network, population-size encoding as well as population-size as policy input. It is shown that PAPO outperforms other benchmark algorithms in terms of metrics such as NashConv, and the generated policies from PAPO are also used to obtain some observations and understandings of the convergence behavior of policies (i.e., scaling laws) with an attempt to further deepen the understanding of the connection between N-player games and mean-field games.

**Summary Of The Review:**

Given the above comments, I think this paper is generally a well-written paper with some notable novelties. Hence I would like to recommend this paper for acceptance.

---

> ### Author Response · Authors · 2022-11-18
> **Response to Reviewer cUv7**
>
> Thanks for your valuable comments and your appreciation that our work has notable novelties. We summarize the questions and present our responses below.
>
> ---
> **Q1. Too much approximation/inaccuracies in the study of the scaling laws, making it not very convincing.**
> * We agree with the reviewer that our results are approximations. In fact, the most exact/accurate approach is to obtain the closed-form solution of each game and then study the evolution of the solutions. But computing the exact solution for a multi-player general-sum game is typically difficult. Thus, training approximate equilibrium policies is a common practice in the literature.
> * There are two aspects to show that the inaccuracies might have been kept as minimal as possible in our work (note that it could be more involved to rigorously control the inaccuracies). (1) As shown in Appendix D.2, Figure 6 and Figure 7, the training curves and BR training curves have approximately converged after a sufficiently large number of training episodes. In this sense, we could expect that our method would return the approximate Nash/BR policies (i.e., the policies have low approximate NashConv). (2) A clear trend of the evolution of the generated policies (i.e., scaling laws) has been observed (Figure 3, Panel B), which, to some extent, also shows that the generated policies are probably approximate Nash policies. This is because the premise to investigate the scaling laws of the policies is that they are indeed (approximate) Nash policies.
> * In the above sense, an important step is to propose efficient algorithms to generate efficient policies (approximate Nash policies) for the games with different population sizes before further investigating the scaling laws of the policies, which is one of the contributions of this work.
>
> ---
> **Q2. Concerns about Proposition A.1.1.**
> * We thank the reviewer for pointing out the gap in Proposition A.1.1. Indeed, without extra conditions, we cannot directly achieve the conclusion that $G(N)$ is an MPG. We fix this issue and restate the proposition to include the conditions for deriving the conclusion. As the reviewer mentioned, this is not the focus of this work. We try to establish some connection between our work and the most recent advances in the convergence of independent policy gradient.
>
> ---
> **Q3. The generalization across different games with different number of players is novel, but I think the authors are over-claiming their contribution in terms of studying the scaling laws. ...**
> * We agree with the reviewer that many works have tried to study the connection between the MFG and the corresponding finite-agent games from a theoretical or computational viewpoint. In the revision, we reorganize the Related Work section to position our work within the literature more accurately. In this sense, we note that it is a misunderstanding that our contribution is over-claimed. Meanwhile, the Abstract and Introduction are also properly modified to show the difference between our work and existing works.
> * Overall, there are mainly two aspects that differentiate our work from existing works in terms of studying the scaling laws. (1) Most of the existing works consider the convergence of the empirical distribution of the population as the number of players goes to infinity, or derive the general result that the NE of an MFG is an approximate NE of the corresponding finite-agent game (as pointed out in [1]). In contrast, our work aims to establish the connection between the MFG and finite-agent games from an agent-centric perspective, i.e., by directly investigating the evolution of the optimal policies of agents as the number of agents increases (from finite to infinite). (2) Recently, due to their powerful expressiveness, deep neural networks have been widely used to represent the policies of agents in both finite-agent games and MFGs (which, as the reviewer mentioned, is more general). In this sense, our work is the first attempt to establish the connection between the MFG and the corresponding finite-agent MGs.
>
> ---
> **Q4. The reproducibility looks good in general, but the detailed hyper-network structure in Appendix B.1 might better be made clearer with an illustration figure, and the shapes of $x$, $\boldsymbol{w}_2(z)$, $\boldsymbol{g}_2(z)$ and $\boldsymbol{b}_2(z)$ and why they match might better be made clearer as well.**
> * Thanks for the reviewer's suggestion. In the revision, we use a figure (Figure 6) to illustrate the structure of the hypernetwork. In addition, in Figure 7, we show the code snippet in the forward process of PAPO to more intuitively show how the shapes of $x$, $\boldsymbol{w}_2(z)$, $\boldsymbol{g}_2(z)$ and $\boldsymbol{b}_2(z)$ match.
>
> ---
> [1] Xin Guo and Renyuan Xu. Stochastic games for fuel followers problem: N versus MFG. SIAM Journal on Control and Optimization, 57(1):659–692, 2019.

---

> > ### Comment · Reviewer_cUv7 · 2022-11-27
> > **Response to the authors' responses**
> >
> > I sincerely thank the authors for the careful revision of the draft and the detailed reply. Many of my concerns have been addressed, and particularly, I'm glad that the authors have adopted many of the suggestions to improve the draft, like Figure 7 and fixing the error related to potential games in Appendix A. But since I have already given a score of 8, I decide to maintain my score. That being said, for the reference of the (senior) area chairs and program chairs, I would like to confirm again that the overall quality of the current draft has been obviously further improved.

---

> > > ### Author Response · Authors · 2022-11-27
> > > **Thanks for your feedback**
> > >
> > > Dear Reviewer cUv7,
> > >
> > > Thanks for your feedback and your high score which supports our work. We are glad that our responses and revised draft have addressed your concerns. We are always welcome for further discussion if you have any concerns.
> > >
> > > Sincerely,
> > >
> > > Authors of Paper3814

---

### Author Response · Authors · 2022-11-18
**To All Reviewers**

We thank all the reviewers for their valuable and insightful comments. We have uploaded a new version and the major changes have been highlighted in blue. We summarize them as follows.

* We reorganize the Related Work section to more accurately position our work within the literature, which avoids the misunderstanding that our contribution is over-claimed. The Abstract and Introduction are also properly modified accordingly.

* We add more details in Appendix B.1 and C.2 to further improve the reproducibility of our work.

* We add more explanations on the experimental results in Appendix D.1. For completeness, we add the evaluation results of PAPO w/ RE and more explanations in Appendix D.4.

* We restate some statements, e.g., the effect of the population size and Proposition A.1.1, such that they are more rigorous.

* We fix some typos.

---

### Decision · Program_Chairs · 2023-01-20

**Decision:**

Accept: poster

**Justification For Why Not Higher Score:**

It's not very clear if the result of this paper is significant/ will lead to practical impact. The presentation needs to be improved,

**Justification For Why Not Lower Score:**

The results are new and interesting. The experiments are solid.

**Metareview: Summary, Strengths And Weaknesses:**

This paper studies the scaling laws of mean-field games by proposing a variant of independent proximal policy optimization (PPO), called Population-size-Aware Policy Optimization (PAPO), that trains Nash equilibrium agent policies that generalize across different number of agents in homogeneous games by utilizing hyper-network, population-size encoding as well as population-size as policy input. It is shown that PAPO outperforms other benchmark algorithms in terms of metrics such as NashConv, and the generated policies from PAPO are also used to obtain some observations and understandings of the convergence behavior of policies (i.e., scaling laws) with an attempt to further deepen the understanding of the connection between N-player games and mean-field games.

During the AC-reviewer meeting, reviewers are convinced with the part this paper considers a relatively new and very interesting problem, present good empirical results, and thus make solid contributions. Therefore we recommend acceptance. However, there are still some concerns we hope authors can address in the final version. (1) the title reads misleading. Multiple reviewers have pointed this out, that the word "scaling laws" typically mean things that are very different from the definitions in this paper. We would suggest to use a different word, instead of redefining "scaling laws" at the beginning of the abstract; (2) the author may want to consider the rigorous math definition of "scaling laws" (or actually the policy change according to the number of player). In particular, whether the Nash equilibria is unique in games discussed in this paper. If it's not unique, how to properly define the policy change/convergence when population size changes (i.e., which Nash in the new game should be comparing to). Since this is an empirical paper, we evaluate this paper mostly from the novelty and performance from the empirical perspective. Nevertheless, answering these basic questions rigorously can greatly improve the quality of this paper.

**Note From Pc:**

if the above contains the word "oral" or "spotlight" please see: "oral" presentation means -> notable-top-5% and "spotlight" means -> notable-top-25%. As stated in our emails, we are disassociating presentation type from AC recommendations

**Summary Of Ac-Reviewer Meeting:**

Strength:
1. (30%) Question is new and interesting, this paper makes solid progress for the field.
2. (30%) The experiments are correctly set up. The added discussion in revision resolves reviewers’ concerns.

Weakness:
1. (15%) Writing / presentation: the title is particularly misleading, the word "scaling laws" does not match their standard meaning.
2. (15%) The practical impact of this result remain unclear.
3. (10%) Overall framework may lack rigor or theoretical justification.